**Global soil-climate-biome diagram: linking surface soil properties to climate and biota**

Xia Zhao[1], Yuanhe Yang[1], Haihua Shen[1], Xiaoqing Geng[1], Jingyun Fang[1,2]*

[1]State Key Laboratory of Vegetation and Environmental Change, Institute of Botany, Chinese Academy of Sciences, Beijing, China 100093

[2]College of Urban and Environmental Sciences, and Key Laboratory for Earth Surface Processes of the Ministry of Education, Peking University, Beijing, China 100871

*Correspondence to*: Jingyun Fang (jyfang@urban.pku.edu.cn)

**Abstract.** Surface soils interact strongly with both climate and biota and provide fundamental ecosystem services that maintain food, climate, and human security. However, the quantitative linkages between soil properties, climate, and biota

remain unclear at the global scale. By compiling a comprehensive global soil database, we mapped eight major soil properties (bulk density; clay, silt, and sand fractions; soil pH; soil organic carbon [SOC] density; soil total nitrogen [STN] density; and soil C:N mass ratios) in the surface soil layer (0-30 cm) based on machine learning algorithms, and demonstrated the quantitative linkages between surface soil properties, climate, and biota at the global scale, which we call the global soil-climate-biome diagram. On the diagram, bulk density increased significantly with higher mean annual

temperature (MAT) and lower mean annual precipitation (MAP); soil clay fraction increased significantly with higher MAT and MAP; Soil pH decreased with higher MAP and lower MAT, and the 'critical MAP', which means the corresponding MAP at a soil pH of =7.0 (a shift from alkaline to acidic soil), decreased with lower MAT; SOC density and STN density both were jointly affected by MAT and MAP, showing an increase at lower MAT and a saturation towards higher MAP. Surface soil physical and chemical properties also showed remarkable variations across biomes. The soil-climate-biome

diagram suggests the shifts in soil properties under global climate and land cover change.

## 1. Introduction

As a critical component of the Earth system, soils influence many ecological processes that provide fundamental ecosystem services (Amundson et al., 2015; Milne et al., 2015; Adhikari and Hartemink, 2016). Soil physical properties, such as bulk density and soil texture, are important for water retention and the preservation of carbon (C) and nutrients (Hassink, 1997;

Sposito et al., 1999; Castellano and Kaye, 2009; Stockmann et al., 2013; Jilling et al.,2018), whereas soil chemical properties, such as soil acidity (pH), organic C, and nutrient contents, are essential regulators of nutrient availability and plant growth, further affecting C and nutrient cycling as well as vegetation-climate feedbacks (Davidson and Janssens, 2006; Chapin et al., 2009; Milne et al., 2015). As the most biogeochemically active soil layer, surface soil dominates the soil function and interacts strongly with climate and vegetation (Jenny, 1941; Alexander, 2013; Weil and Brady, 2016). Therefore, assessing

the physical and chemical properties in surface soil could provide insights of global soil functions and support soil stewardship to secure sustainable ecosystem services (Batjes, 2009; Sanchez et al., 2009; Koch et al., 2013).

In the context of rapid environmental change, there is an increasing need for high-quality, high-resolution, and timely updated global mapping of soil properties (Grunwald et al., 2011). Based on the global database of soil properties (e.g., the Harmonized World Soil Database [HWSD]), multiple linear regression models have been widely used for soil mapping (Batjes 2009; Hengl et al. 2014). Although recent progress has been made by compiling larger numbers of soil profiles and performing accuracy assessments, the maps of global soil properties are subject to weak relationships between soil properties and the corresponding predictors (Hengl et al., 2014). Moreover, some attempts have been made to predict global soil properties based on Earth system models, but these predictions frequently showed large variation among different models and agreed poorly with observational data (Todd-Brown et al., 2013; Tian et al., 2015). Recently, machine learning algorithms, such as random forest (RF) analyses have been successfully applied to develop spatially explicit estimates of soil organic C (SOC) (Grimm et al., 2008; Wiesmeier et al., 2011; Ding et al., 2016; Hengl et al., 2017). Compared with multiple linear regression models, RF analysis has several advantages, such as the ability to model non-linear relationships, handle both categorical and continuous predictors, and resist overfitting and noise features (Breiman, 2001).

The underlying stability of soil systems is controlled by their inherent balance between mass inputs and losses, which strongly feeds back on climate and biota (Amundson et al., 2015; Weil and Brady, 2016). By overlapping the spatial distribution of climate types, biome types, and soil orders, Rohli et al. (2015) first quantified the percentage of global land surface that is covered by the combinations of climate types, biomes, and soil orders. However, quantitative linkages of soil properties, climate, and biota have not yet been developed in a common diagram. Encouragingly, significant progress in digital soil mapping techniques and the rapidly growing quantity of recorded soil information (Sanchez et al., 2009; Grunwald et al., 2011; Arrouays et al., 2014; Hengl et al., 2014; Shangguan et al., 2014), provide a great opportunity to assess the quantitative linkages between soil properties, climate, and biota at the global scale.

In this study, we first compiled a global soil database (GSD, see Materials and Methods) that contains more than 28000 soil profiles for seven soil physical and chemical properties in surface soil layer (0-30 cm), including bulk density (g cm$^{-3}$), sand, silt and clay fractions (%), soil pH, SOC density (kg m$^{-2}$), and soil total nitrogen (STN) density (kg m$^{-2}$). Using regional RF algorithms, we then established global soil maps for eight soil properties (the above mentioned seven soil properties plus C:N ratios, being estimated based on SOC density and STN density) at a 1-km resolution and evaluated their corresponding uncertainties. On the basis of Whittaker biome diagram which illustrates the essential role of climate in shaping the spatial pattern of global biomes (Whittaker, 1962), we further developed a global soil-climate-biome diagram by plotting each soil property on a climate basis as climate and vegetation are two key soil-forming factors (Jenny, 1941). Although parent material (e.g., bedrock) also plays an important role in affecting soil properties, it affects soil formation at a relatively long time scale (Chesworth, 1973), particularly in the subsoil (Gentsch et al., 2018). In addition, our soil-climate-biome diagram thus focuses on soil properties in the surface layer, given that surface soils are dynamic in time and likely interacting more instantly with climate and vegetation than deeper soils (Weil et al., 2016). Overall, our objectives were to (i) map the

physical and chemical properties of global surface soils, and (ii) determine the linkages between surface soil properties, climate and biota at the global scale.

## 2. Materials and Methods

### 2.1 Data set

We compiled ground-truth soil property data to establish a comprehensive database of worldwide soil profile information (Global soil database, GSD). Our GSD includes existing sources of soil profile data from the International Soil Reference and Information Centre-World Inventory of Soil Emission (ISRIC-WISE) Potential database (version 3.2; Batjes, 2009), soil reference profiles of Canada (Pan et al., 2011), the Land Resources of Russia/International Institute for Applied Systems Analysis (IIASA) (http://nsidc.org/data/ggd601.html), the International Soil Carbon Network (ISCN 2012,
http://www.fluxdata.org/nscn/Data/AccessData/SitePages/Carbonto1M.aspx), the Soil Profile Analytical Database of Europe (SPADE), the Northern Circumpolar Soil Carbon Database (NCSCD, Tarnocai et al., 2009), the Second State Soil Survey of China (National Soil Survey Office, 1998), literature-retrieved soil data on the forests of China (Yang et al., 2014), field campaign data on the grasslands of northern China (from our research team; Yang et al., 2008, 2010), and field survey data of Australia (Wynn et al., 2006) (see Table S1 for more detailed information on these data sources). Overall, the GSD
includes more than 28000 soil profiles (Fig. 1; Table S1). Although the total sample number and spatial distribution of the profile data are similar to those of the WISE30sec (Batjes, 2016), the GSD includes more specific soil data from China. Nonetheless, both databases include limited profiles for some regions of the world, notably Australia, Sahara and the northern territories of both Canada and Russia (Fig. 1).

The GSD includes field measured data of four soil physical properties (bulk density [g cm$^{-3}$], and sand, silt and clay fractions [%]), three chemical properties (soil pH, SOC density [kg C m$^{-2}$], and STN density [kg N m$^{-2}$]) in the surface soil layer (see Table S2 and Fig. S1 for more details), and general information on soil sampling (site location, sampling time, and data source). Data harmonization was conducted by three steps. First, we screened sampling and measurement approaches of each soil property and excluded data those were not comparable to others in methodology. For instance, geographic coordinate
data were included only when WGS84 or a geographic coordinate system that could be converted to WGS84 projection was used; Soil texture data were included only when the internationally accepted particle size class were used (clay < 2 μm < silt < 50 μm < sand < 2000 μm). This allowed us to construct a database of soil properties with comparable methodology. Second, we excluded records with no information on the target soil depth (0-30cm). In case that soil organic matter was measured instead of SOC, we used a Bemmelen index (0.58) to convert soil organic matter into SOC. If data of bulk density
were not provided, we estimated them based on regional-specific pedotransfer functions (Schaap and Leij, 1998; Yang et al., 2007; Abdelbaki, 2018) (Table S3). Specifically, we established empirical relationship between bulk density and SOC content to estimate bulk density based on measured SOC for those cases with missing data of bulk density. There were 42%

profiles with measured data on bulk density and 58% profiles with estimated data on bulk density. It is true that the correction for rock fragment is important for the estimation of soil C stocks, but it remains a global challenge because existing databases usually contain limited information on gravel fractions (Jandl et al., 2014). Nevertheless, the inclusion of gravel has been evidenced to exert a relatively low impact on the calculation of SOC stocks in the surface soil layer (0-30 cm), mainly due to the fact that surface soil usually contains a low proportion of gravels (Saiz et al., 2012). Therefore, we assumed no rock fragment or the rock issue had been handled once it was not reported. Finally, we extracted data on soil properties of the 0-30cm soil depth and calculated the means of each soil property. SOC (STN) density was calculated based on bulk density and SOC (STN) content.

The GSD also contains pedologic information on soil orders and the horizons of the sampled soil profiles, mean annual temperature (MAT), mean annual precipitation (MAP), seasonality of air temperature (TS, calculated as $100 \times SD_{monthly}/Mean_{monthly}$) (Xu & Hutchinson, 2011), seasonality of precipitation (PS), mean annual normalized difference vegetation index (NDVI), elevation (global digital elevation map [DEM]), slope, and land use type for each recorded site (see Table S4 for more details). Notably, it was difficult to harmonize data of soil orders and further quantify their roles, because data on soil orders were originally reported based on several different soil classification systems with different standards (Carter and Bentley, 2016). It was the same case for soil horizon. Additionally, horizon information was not reported in some cases (accounting for 15% profile), while soil depth was well documented in our database. Therefore, we were not able to consider the role of soil horizons and instead we simply estimated the mean soil properties by a fixed depth of 30 cm. Nevertheless, the depth of 0-30 cm has been frequently used in the mapping and modelling of surface soil properties at regional and global scales (e.g., Batjes, 1997; Yang et al., 2010; Saiz et al., 2012; Wieder et al., 2013; Shangguan et al., 2014). As 96% of soil profiles in GSD were sampled during 1950 to 2000, we thus used multiple-year (1950-2010) averages of climatic variables from WorldClim database. For sites with missing reports on climate or topographical data, profile coordinates were used to derive data at each site using a selection of GIS layers, from the WorldClim database for MAT and MAP and GTOP30 DEM-derived surfaces.

## 2.2 Region-specific random forest model

The random forest (RF) model is a data mining algorithm to make predictions based on an ensemble of randomized classification and regression trees (Breiman, 2001). We mapped soil properties based on a region-specific RF approach that yields spatially explicit estimates of each pixel (see Fig. S2 for more details on the workflow of this approach). To overcome spatial biases of the database (for example, heavy sampling in the USA), we divided the global land into 11 regions: Africa, Australia, Canada and Alaska, East Asia, Europe, Mexico, Russia, South America, tropical Asia, the USA, and West Asia (Table S2). In each region, we first constructed a RF model using the regional datasets and then used the model to estimate the spatial distribution of each soil property at a resolution of 1 km. Predictions were based on eight environmental variables, including MAT, MAP, TS, PS, vegetation cover conditions (NDVI), elevation, slope, and land use type (see Table S4 for

more details on the data sources of each variable). To obtain a robust variogram, soil property data below 2.5% quantile and above 97.5% quantile were excluded as outliers and were not used for modeling in each region (Pleijsier, 1989; Jiménez-Muñoz et al., 2015). Notably, these excluded samples were distributed relatively randomly in space (Fig. S3). By conducting an analysis using all data samples and comparing the results with those excluding outliers, we found similar spatial patterns

and means of global surface soil properties (Fig. 2 & Fig. S4) but lower cross-validated $R^2$ when including all samples (Table 6 & Table S5). This implies an improvement of prediction by excluding outlier samples.

Because a large number of regression trees are constructed, one major advantage of RF model is that the risk of overfitting can be reduced. Another advantage is that the prediction depends on only three user-defined parameters: the number of trees

(ntree), the minimum number of data points at each terminal node (nodesize), and the number of features sampled for splitting at each node (mtry). We used ntree = 1000 (default ntree = 500) in order to achieve more stable results (Grimm et al., 2008). For nodesize and mtry, we used the default set for RF regression. Also called a "black box" approach, one major disadvantage of RF model is that the relationships between the response and predictor variables cannot be interpreted individually for every RF tree. The relative importance of variables, denoted by the percent increase in mean-squared error

(%IncMSE), was estimated based on a permuting out-of-bag (OOB) method (Strobl et al., 2009 a & b). For each tree of the random forest, we compared the prediction error on the OOB portion of the data (MSE for regression) with that after permuting each predictor variable. The differences were then averaged over all trees, and normalized by the standard deviation of the differences. The relative percent (mean/SD) increase in MSE as compared to the out-of-bag rate (with all variables intact) was used to indicate the relative importance of each variable (Breiman, 2001).

**2.3 Uncertainty analysis**

In each region, we used 10-fold cross-validation to estimate the average mapping accuracy for each target soil property. The modelling accuracy for each bootstrap sample was evaluated by the amount of variation explained by the models ($R^2$) and by the root mean square error (RMSE) calculated based on the observational and predicted soil property in the independent validation dataset (Table S5). Model uncertainties were assessed based on the bootstrap method. A robust estimate was

derived by averaging the 10-fold cross-validation samples, and the uncertainty of the estimates was calculated as the standard deviation (SD) of the 10-fold cross-validation (Fig. S4).

**2.4 Statistical analysis**

Based on the results of the ensemble models, we mapped each soil property (bulk density, sand, silt, clay, pH, SOC density, and STN density) and their uncertainty at a resolution of 1 km. Soil C:N ratios were mapped based on values of SOCD and

STND, and its uncertainty could be jointly indicated by the uncertainties of SOCD and STND. We also plotted each soil property on a modified Whittaker biome diagram. To explore the roles of MAT and MAP as well as their interactions, we averaged soil property values for each MAT×MAP combination by a division of 1°C×100mm and explored quantitative

linkages between soil properties and climate variables (MAT/MAP) for different climate types (humid *vs.* arid; warm *vs.* cold). Specifically, we used a MAP threshold of 500 mm to differentiate relatively humid versus arid climates (Holdridge, 1967), while a MAT threshold of 10 °C was used to separate relatively warm and cold climates (Trewartha and Horn, 1980). To explore the role of MAT in regulating the critical MAP for a shift from alkaline to acidic soil, we further plotted the critical levels of MAP (100 mm division) at soil pH=7.0 with MAT. We then compared the soil properties across main biomes, including tropical forest, temperate forest, boreal forest, tropical savannahs and grasslands, temperate grasslands and shrublands, tundra, permanent wetlands, deserts and croplands. All statistical analyses were performed using Matlab 2015a (The MathWorks Inc., Natick, MA, USA). Values were presented as mean ± standard deviations, if not specially noted.

## 3. Results

### 3.1 Global mapping of soil properties

Our results agreed well with the observed data across most regions (Fig. S5), and the ensemble models generally explained 30~60% of the variation in soil properties (Table S5). The eight soil properties showed great spatial heterogeneity across the globe in the upper 30-cm layer (Fig. 2). For instance, bulk density showed low values in the northern latitudes in the Eurasian continent, whereas high values occurred in the USA, North Africa, West Asia, and India (Fig. 2a). The clay fraction exhibited lower values at higher latitudes, whereas higher levels of sand fraction occurred at lower latitudes (Figs. 2b, 2c). The pH value of the surface soil was high (generally > 7.0) in arid regions and it was relatively low (generally < 6.0 ) in most forested regions (Fig. 2e). The spatial patterns of SOC density and STN density were generally similar, both showing greater values at higher latitudes in the northern hemisphere and no consistent change with latitude in the southern hemisphere (Figs. 2f, 2g). Specifically, SOCD and STND both showed highest values in the northern high latitudes, while low values occurred in semiarid and desert regions. Soil C:N ratio showed the highest values at high latitudes in northern hemisphere, while the lowest values occurred in arid regions in Northern Africa, West Asia and Southern Europe (Fig. 2h).

### 3.2 Global soil-climate-biome diagram

By placing data of surface soil properties on the Whittaker climate-biome diagram (Whittaker, 1962), we then documented the linkages between soil properties and climate across global biomes. We call this as the global soil-climate-biome diagram (Fig. 3). Specifically, bulk density decreased with lower MAT and higher MAP (Fig. 3a, Figs. 4a, 4b); sand fraction was inversely related to MAP and MAT (Fig. 3b, Figs. 4c, 4d), whereas the clay fraction showed an opposite pattern (Fig. 3c, Figs. 4e, 4f); and soil pH increased with higher MAT in arid climate (MAT ≤ 500 mm) (Figs. 3e, 5a), while it decreased significantly with higher MAP both in cold (MAT ≤ 10°C) and warm (MAT > 10°C) climates (Figs. 3e, 5b). The critical MAP for the transition from alkaline to acidic soil (pH = 7.0) showed a non-linear increase with MAT and reached to a maximum of 400-500 mm when MAT exceeded 10 °C (Fig. 5c).

SOC density in the upper 30-cm soil layer decreased significantly with MAT at both arid (MAT ≤ 500 mm) and humid climates (MAT > 500 mm) (Figs. 3f), whereas it increased with MAP in accordance with a saturation curve (cold climate, MAT ≤ 10°C: SOCD = 0.0737×MAP/(1+0.0049×MAP); warm climate, MAT > 10°C: SOCD = 0.0144×MAP/(1+0.0016×MAP) ), showing a higher saturation threshold in cold climates (14.5 kg C m$^{-2}$) compared with that in warm climates (8.0 kg C m$^{-2}$) (Figs. 6b). Similarly, STN density decreased significantly with MAT (Figs. 6c) and increased with MAP in accordance with a saturation curve (cold climate: STN = 0.0401×MAP/(1+0.0502×MAP); warm climate: STN = 0.0015×MAP/(1+0.0021×MAP) ), showing a higher saturation threshold in cold (0.80 kg N m$^{-2}$) than warm climates(0.65 kg N m$^{-2}$) (Figs. 6d). Combining the trends of SOC density and STN density, the C:N ratio of the upper 30-cm layer increased with MAT at a climate of MAT < 0 °C and then decreased (Fig. 6e). In contrast, the C:N ratio increased with MAP in accordance with a saturation curve (cold climate: C:N = 0.1450×MAP/(1+0.0080×MAP); warm climate: C:N = 0.3781×MAP/(1+0.0308×MAP) ), showing a higher saturation threshold in cold climates (18:1) compared with warm climates (12:1) (Fig. 6f).

Soil properties showed varied values across and within biomes throughout the world (Table 1; Fig. 3). Mean bulk density was lowest in tundra and boreal forest, and it was highest in the desert and tropical thorn scrub and woodland (Tables 1). Mean sand fraction was highest in boreal forest, whereas mean clay fraction was highest in the tropical rainforests (Tables 1). Soil pH was generally lower than 5.5 in tropical forest, boreal forest and tundra, but mean pH values could approach and even exceed 7.0 in dry biomes, such as the desert, grassland and savanna (Table 1). Moreover, means of SOC and STN densities both showed high values in boreal forest and tundra, but they were extremely low in the desert and tropical thorn scrub and woodland (Table 1). Mean soil C:N ratio showed the highest values in tundra and boreal forest (> 15:1), while it was lowest in desert, temperate shrubs and grasslands (≤ 10:1) (Table 1; Figs. 3h). On average, the global means of SOC density and STN density were 6.94 (SD= 4.42) kg C m$^{-2}$ and 0.53 (SD= 0.23) kg N m$^{-2}$ in surface soils, summing up to a global total storage of 797 ± 4.1 Pg C ($10^{15}$ g, or billion tons) and 64 ± 0.4 Pg N, respectively (Table 1).

## 4. Discussion

### 4.1 Linkages between climate and surface soil physical properties

The soil-climate-biome diagram demonstrated the quantitative linkages between surface soil physical properties and climate variables at the global scale. Compared with variables associated with topography (e.g., elevation and slope), vegetation activity (i.e., NDVI) and land cover (i.e., land use type), climate variables (such as MAT, MAP, TS and PS) were stronger predictors of bulk density and soil texture (Fig. 7a-c). This was likely due to the essential role of temperature and precipitation in physical, chemical and biological processes during soil formation (Weil and Brady, 2016). Specifically, bulk density showed an increase with higher MAT and lower MAP, likely due to an accompanying decrease of SOCD

(Ruehlmann and Körschens, 2009) which was driven by stronger microbial decomposition under warmer and wetter conditions (Fig. 6; see more discussion on the effect of climate on SOCD in section 4.3; Wiesmeier et al., 2019). In addition, higher MAT and MAP can accelerate the rate of weathering (Jenny, 1941; Lal, 2018), thus resulting in lower sand fraction and higher soil clay fraction (Fig. 4). Along with topographical variables, climate may also affect soil physical properties via erosion processes. For example, soil erosion is highly selective to silt, while sand is less mobile due to high weight and clay is protected by soil aggregates (Wischmeier and Mannering, 1969; Torry et al., 1997; Wang et al., 2013).

Other factors, such as historical tectonics, glaciations and soil ages, could also affect soil physical properties (Jenny, 1941; Weil and Brady, 2016), but they are often spatially correlated with climate variables, making it difficult to separate their role from the latter. For instance, the effect of glaciations is stronger, the soil age is younger and air temperature is lower towards higher latitudes. Likewise, the role of tectonics in rejuvenating younger soils might also be mixed by corresponding climatic conditions across altitudinal gradients. In tropical regions, we found a significant decrease in bulk density and clay fraction with higher elevation (Fig. S6a&d). This decrease of bulk density along the altitude gradient was likely due to an increase in SOC retention (Fig. S6f), being resulted from low rates of soil organic matter decomposition along with lower temperature (Grieve et al., 1990; Kramer and Chadwick, 2016). Meanwhile, the decrease of clay fraction with higher altitude was likely due to a younger soil age (Waite and Sack, 2011), lower weathering rate under lower temperature (Grieve et al., 1990; Kramer and Chadwick, 2016), and a downslope translocation of surface soil to lower altitude. Interestingly, these altitudinal gradients were consistent with the results of field studies (Dieleman et al., 2013) and also mirrored a similar trend across latitudes.

## 4.2 Key role of climate in determining global patterns of surface soil chemical properties

Our results indicated that MAP was the most important surrogate for soil pH prediction (Fig. 7d). Such a pattern might be due to the increased leaching of exchangeable base cations across large-scale precipitation gradients (Jenny et al., 1941). Interestingly, further analysis showed that the critical levels of MAP for the transition from alkaline to acidic soil decreased non-linearly with lower MAT owing to changing water balance (Fig. 5). Specifically, the critical MAP ranged from 400-500 mm when the MAT exceeded 10 °C and could decrease to 50-100 mm when MAT was close to 0 °C, highlighting significant interactions between MAP and MAT. Such a pattern was supported by a recent study, which revealed that the transition from alkaline to acidic soil occurred when the MAP began to exceed the mean annual potential evapotranspiration (Slessarev et al., 2016). It should be noted that, other factors besides climate variables, such as acid deposition may also contribute to regional-scale patterns of soil pH, especially in Europe, eastern North America and southern China, where have received high-level acid deposition (Bouwman et al., 2002; Vet et al., 2014).

Our analysis also indicated that climate variables (e.g., MAT, MAP) were the strongest predictors of SOC density (Fig. 7e), being in agreement with the findings of previous studies (Post et al., 1982; Gray et al., 2009). Such a pattern reflects the fact

that soil C stock depends on the balance between plant inputs (i.e., litterfall and other plant debris) and microbially mediated metabolic losses of $CO_2$ to atmosphere (Stockmann et al., 2013), which are strongly controlled by climate (Davidson and Janssens, 2006; Bond-Lamberty and Thomson, 2010). In general, precipitation favours net primary productivity (Del Grosso et al. 2008) and the consequent C inputs into the soil, while it intensifies weathering of the parent material and soil acidification, thus increasing formation of SOC-stabilizing minerals (Chaplot et al., 2010; Doetterl et al., 2015) and reducing decomposition of soil organic matter (Meier and Leuschner, 2010). These processes could then explain the increase of SOCD with MAP (Fig. 6), while it did not exceed a certain threshold because of a constraint of C inputs (Del Grosso et al., 2008). Compared with precipitation, temperature largely affects the rate and degree of microbial decomposition of soil organic matter (Wiesmeier et al., 2019). Consequently, SOCD increased with lower MAT (Fig. 6), while it reached saturation due to a threshold of SOM stabilization (Doetterl et al., 2015).

Further analysis also revealed an interaction between MAT and MAP in shaping the patterns of SOC density. For instance, SOC density showed a tendency of saturation with higher MAP, while the saturation thresholds were higher under MAT $\leq$ 10 °C compared with MAT > 10 °C (Fig. 6). Specifically, the saturation threshold for SOC density under MAT $\leq$ 10 °C (14.5 kg C $m^{-2}$) were nearly twice of that under MAT > 10 °C (8.0 kg C $m^{-2}$) (Fig. 6b). These critical levels of SOCD imply a saturation threshold of SOC stocks under certain climate regime (Stewart et al., 2007). Soil C saturation has also been evidenced by experimental studies, which indicate that SOC pool has an upper limit with respect to C input levels because of a threshold of SOM stabilization efficiency (Stewart et al., 2008; Kimetu et al., 2009). These thresholds of soil C saturation can help to estimate soil C sequestration potential and provide important guidelines for regional soil steward and ecosystem management.

Previous meta-analyses indicated that C:N ratio in the soil was well-constrained at the global scale (Cleveland and Liptzin, 2007). Accordingly, our results indicated a strong correlation between STN density and SOC density (Fig. 8) and demonstrated a similar pattern of STN density as SOC density across biome and climate regimes (Figs. 3 & 6). Based on a synthesis of long-term experimental results, Manzoni et al. (2008) demonstrated that the C:N ratio of the litter decreased throughout decomposition. Because soil organic matter is a result of long-term decomposition, surface soil C:N ratio is thus negatively correlated with decomposition degree while positively correlated with SOC content and turnover time (Carvalhais et al., 2014). Our analysis also indicated that higher soil C:N ratios was associated with higher SOC density (Figs. 2, 3).

### 4.3 Shifts in soil properties across biomes and land use types

Our analysis indicated that soil properties varied significantly across global biomes (Table 1). For example, SOC density showed high values in boreal forests and tundra due to the slower microbial decomposition compared with biomass inputs (Hobbie et al., 2000; Hashimoto et al., 2015; Bloom et al., 2016), but these values were extremely low in drylands due to low plant cover and productivity (Delgado-Baquerizo et al., 2013). Due to fast turnover with rapid decomposition of organic

matter, SOC content is relatively poor in tropical forests (e.g., Congo and Amazon tropical forests in Fig. 2f) (Carvalhais et al., 2014; Wang et al., 2018). Accordingly, previous mappings of SOC density have also shown relatively low values in tropical forests (Köchy et al., 2015; Jackson et al., 2017). In view of a strong and negative correlation between SOC and bulk density, bulk density showed an opposite shifts across biomes (Table 1). Moreover, we also found an increase in SOC density and a decrease of soil bulk density with elevation (Fig. S6a & Fig. S6f), likely due to a shift in climate regime and vegetation type.

The effect of land use is important for the SOC stock at a regional to local scale. A change of forest or grassland to croplands can significantly decrease SOC density and thus decrease soil bulk density, while reforestation generally increases SOC density and decreases soil bulk density (DeGryze et al., 2004; Machmuller et al., 2015). When comparing values at the same region (e.g., Southeast Asia), SOC density is obviously lower in croplands than in forests (Fig. S7a and Fig. S7b). This difference has been also evidenced by meta-analysis based on field observations (Don et al., 2011). In the Mediterranean region, an increase in the area of olive plantation and vineyard in last decades have likely contributed to a consequent increase of SOC density (Parras-Alcántara et al., 2013). Moreover, a recent assessment indicates that ecological restoration projects (e.g., Three-North Shelter Forest Program, Natural Forest Protection Project, Grain for Green Program, Returning GrazingLand to Grassland Project) in China has substantially increased soil and biomass C storage in the corresponding regions (Lu et al. 2018). However, our static mapping of global soil properties is not able to account for the effect of temporal land use change on SOC density.

### 4.4 Global carbon and nitrogen stocks in surface soils

Earlier estimates of global SOC and STN stocks were based on either an area-weighted extrapolation or an empirical model of the soil profile data according to climate, vegetation type or soil order (Post et al. 1982, Batjes, 1996; Batjes 2009; Hengl et al. 2014; Scharlemann et al. 2014). In the range of these estimates, our results based on RF modeling indicated that the global stocks of SOC were $788 \pm 39.4$ Pg in the upper 30-cm soil layer. Reports of global STN stocks are relatively rare compared with those of SOC stocks. Based on information on measured soil profiles, Batjes (1996) estimated global STN stocks to be 63-67 Pg N in the upper 30-cm layers. Similar to the estimates by Batjes (1996), our results indicate that global STN stocks were $63 \pm 3.3$ Pg in the upper 30-cm soil layer (Table 1). Despite similar estimation of global total SOC and STN stocks, our regional RF analysis has several advantages, such as the ability to model non-linear relationships, handle both categorical and continuous predictors, and resist overfitting and noise features (Breiman, 2001). By using climate, vegetation, topography and land use variables as predictors, our region-specific RF approach likely produces more robust global maps of soil properties at a finer spatial resolution.

## 4.5 Uncertainties in mapping surface soil properties at the global scale

In this study, we used machine learning algorithms to map global surface soil properties at a 1-km resolution. Although this approach could overcome uncertainties derived from large variations in mapping unit, several limitations still exist in our analysis. First, the limited sample size in certain area may lead to estimation uncertainties. Particularly, the accuracy of the region-specific RF model partially depends on the number of sampling sites and the evenness of the spatial pattern. The limited number and uneven distribution of the soil profile may thus constrain the accuracy of region-specified RF models, especially in regions such as Russia and South America (Table S5).

Second, soil properties have been measured using various approaches and compiled for several decades, while there are no straightforward solutions to accurately harmonize the data at the global level (Maire et al., 2015; Batjes, 2016). The errors due to varied sampling and measurement methods over time may lead to uncertainties in our analysis. Moreover, our data base includes pedologic information on soil orders and soil horizons of sampled soil profiles and these data were originally reported based on several different soil classification systems using different standards (Batjes et al., 2007; Carter and Bentley, 2016). It is thus a challenge for us to harmonize the data on soil orders and soil horizons and quantify their impacts on surface soil properties. Nevertheless, the depth of 0-30 cm has been frequently used in the mapping and modelling of surface soil properties at regional and global scales (e.g., Batjes, 1997; Yang et al., 2010; Saiz et al., 2012; Wieder et al., 2013; Shangguan et al., 2014). By considering essential climatic variables (MAT, MAP, seasonality of air temperature, seasonality of precipitation), vegetation parameters (mean annual NDVI, and land use type), and topographic factors (elevation, slope) that are key to soil formation (Jenny, 1941), our Random Forest analysis may have partially constrained the uncertainties due to the lack of information on soil orders and associated soil horizons.

Finally, uncertainties may also arise from the limited independent variables used in this study. Although essential surrogate variables of climate, topography, vegetation activity, and land cover were incorporated in our analysis, we still could not account for the role of soil horizons, soil ages and parental material characteristics due to the lack of global-scale dataset. For instance, surface soil (top 30 cm) can contain either a single horizon or several very different horizons with very different physical and chemical properties. Soil mineralogy, being a function of parent material, climate and soil age (Jenny, 1941), has been demonstrated to be important in determining the quantity of SOC storage and its turnover time during long-term soil development (Torn et al., 1997). Soil age may also play an important role in forming soil property (Jenny, 1941), but it is hard to evaluate its individual role in regulating spatial patterns of soil properties due to its strong interactions with climate variables. Therefore, future studies should make more efforts to consider these variables when predicting spatial patterns of soil physical and chemical properties at the global scale.

## 5. Conclusion

By compiling a comprehensive global soil database, we mapped eight surface soil properties based on machine learning algorithms and assessed the quantitative linkages between soil properties, climate, and biota at the global scale. Our region-specific random forest model generated high-resolution (1km) predictions of surface soil properties, which can be potentially used as inputs for regional and global biogeochemical models. Our results also produced a global soil-climate-biome diagram, which indicates the quantitative linkages between soil, climate, and biomes. Given that significant changes in major soil properties may occur in view of global environmental change (Trumbore and Czimczik, 2008; Chapin et al., 2009; Todd-Brown et al., 2013; Luo et al., 2016, 2017), more efforts should be made in future to understand the dynamics of the global soil-climate-biome diagram.

## Acknowledgement

This work is supported by National Key Research and Development Program (2017YFC0503901), the National Natural Science Foundation of China (31321061 and 31330012) and Key Research Program of Frontier Sciences, CAS (QYZDY-SSW-SMC011). All data will be available online as a supplement when this paper is published.

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

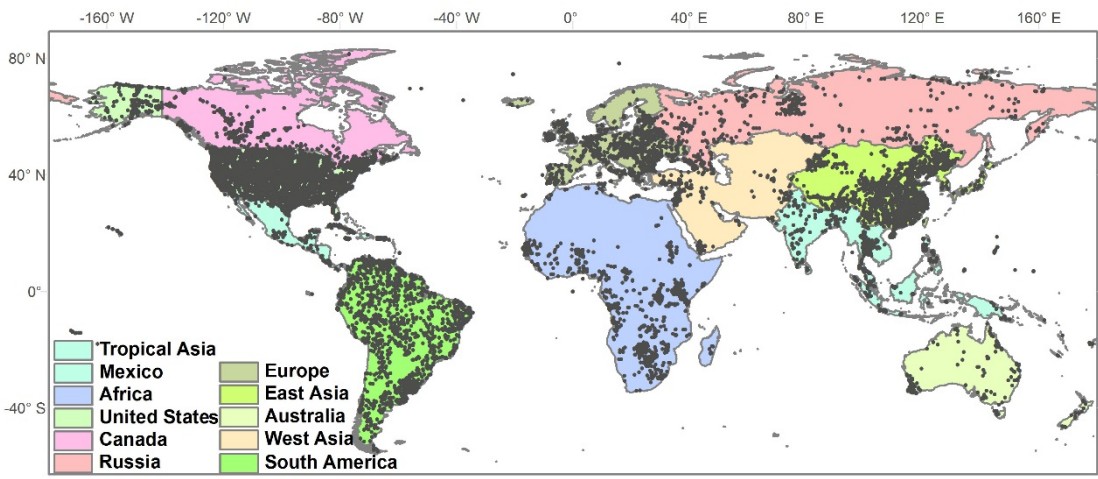

Figure 1. Global distribution of 28222 soil profiles included in the global soil database (GSD).

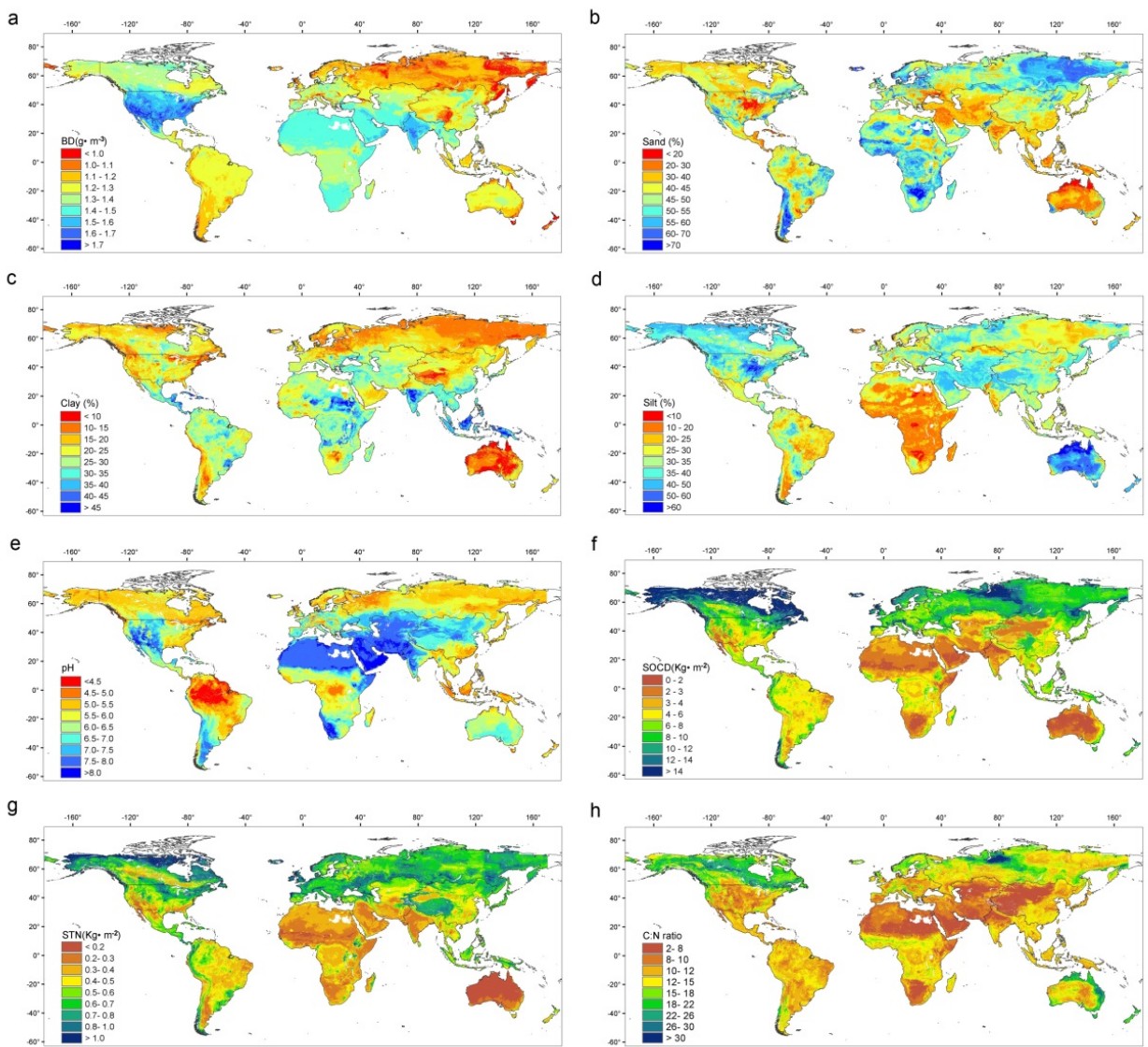

Figure 2. Maps of surface (0-30cm) soil properties. a, BD (bulk density, g m$^{-3}$); b, Sand fraction (%); c, Silt fraction (%); d, Clay fraction (%); e, pH; f, SOCD (soil organic carbon density, kg m$^{-2}$); g, STND (soil total nitrogen density, kg m$^{-2}$); and h, C:N ratio.

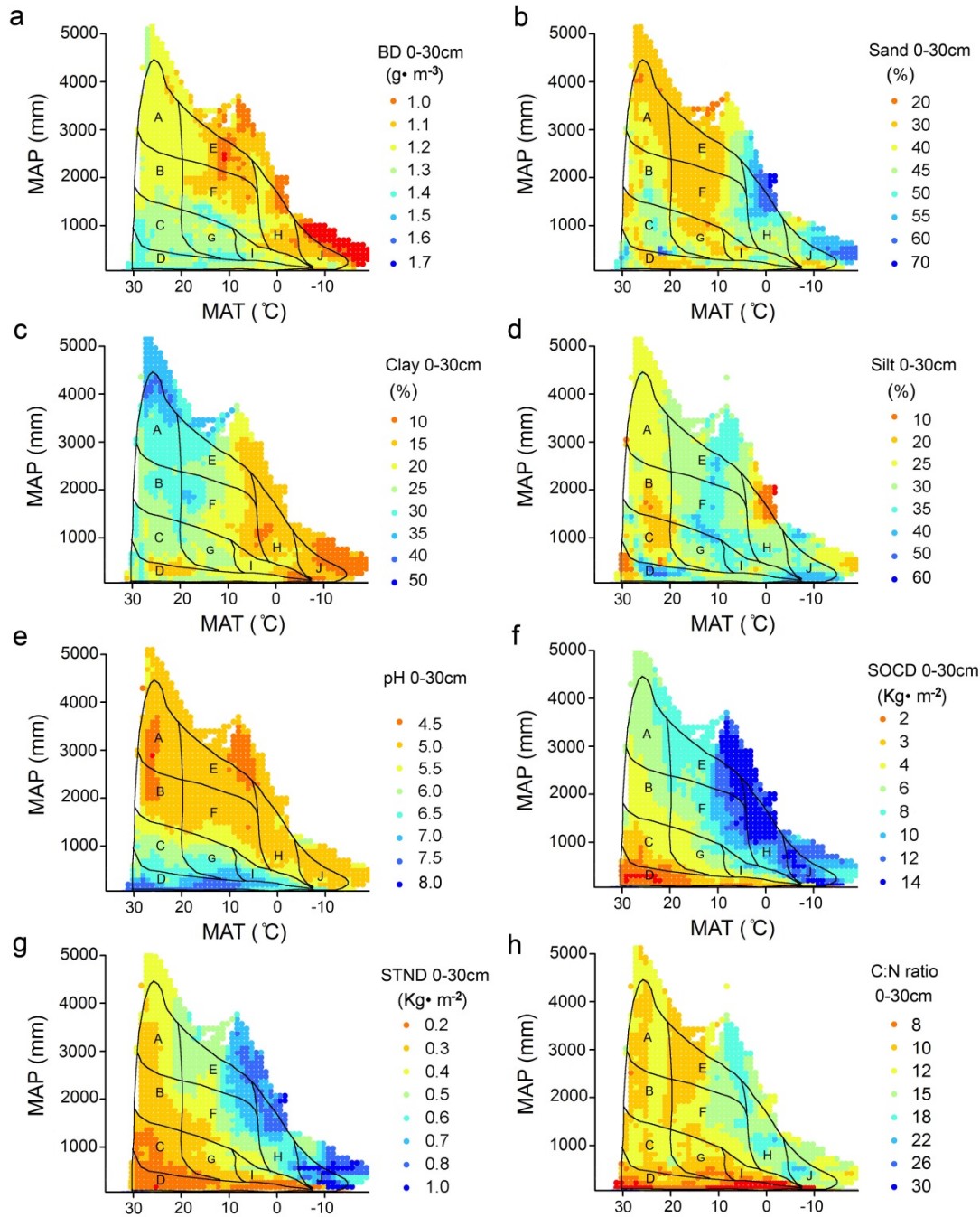

Figure 3. Changes in surface (0-30m) soil properties on the Whittaker biome diagram. Each square shows the average C density within each 1 °C of MAT and 100 mm of MAP. Each biome type in the modified Whittaker biome diagram is indicated by a capital letter. A, Tropical rainforest; B, Tropical seasonal forest; C, Tropical thorn scrub and woodland; D, Desert; E, Temperate rainforest; F, Temperate forest; G, Savanna; H, Boreal forest; I, Grassland; and J, Tundra.

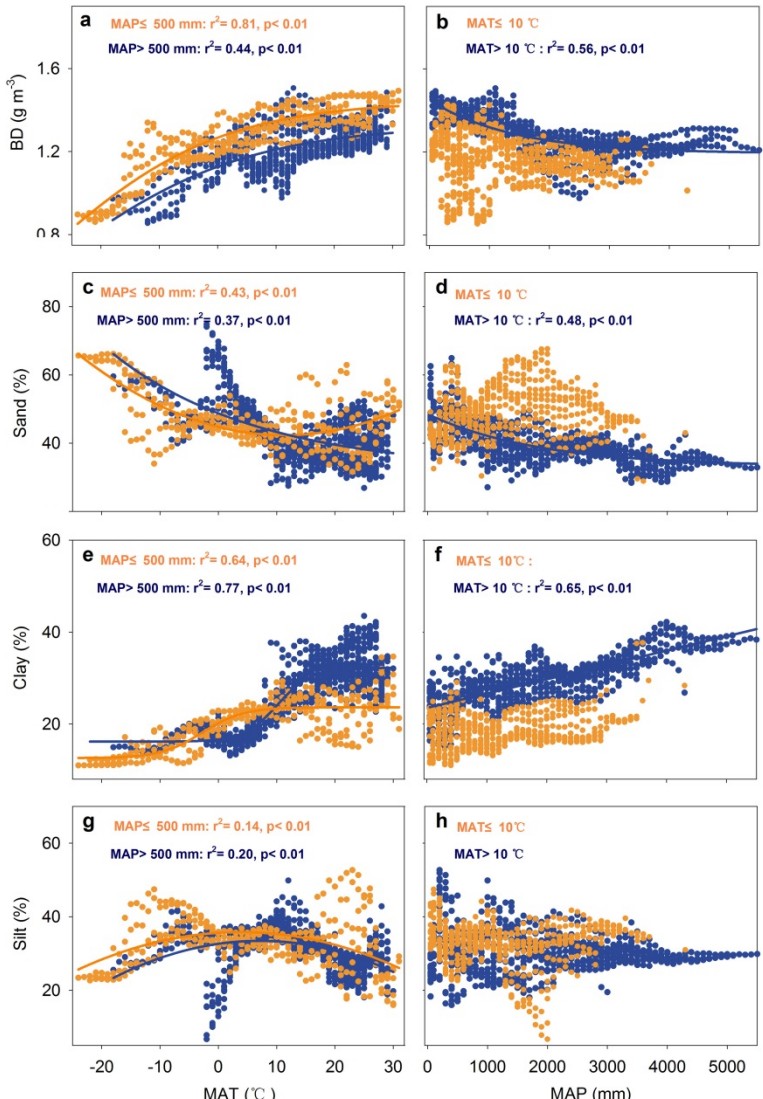

Figure 4. Changes in surface(0-30 cm) soil bulk density (BD, g·m$^{-3}$), sand fraction (%), clay fraction (%) and silt fraction (%) with mean annual precipitation (MAP) and mean annual temperature (MAT). We used 500 mm of MAP as a threshold of transition from arid to humid climate, and 10 °C of MAT as a threshold of transition from cool to warm climate.

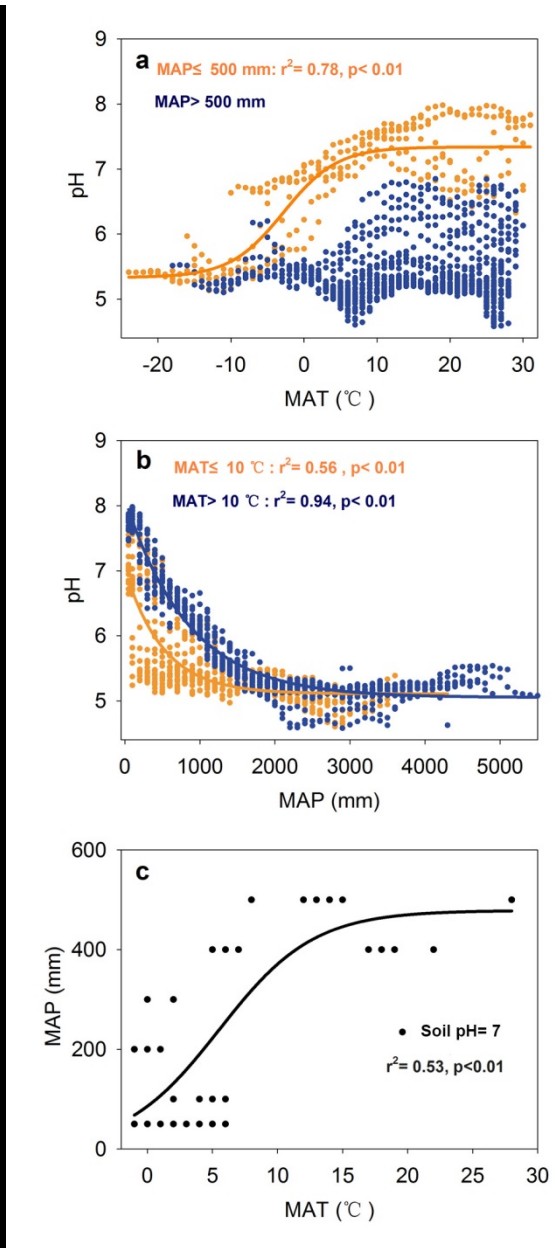

Figure 5. Changes in surface(0-30 cm) soil pH with climate. a, mean annual temperature (MAT); b, mean annual precipitation (MAP); and c, changes in critical of MAP at soil pH=7.0 with MAT. The 'critical MAP' here means the corresponding MAP at a soil pH of=7.0, which indicates a shift from alkaline to acidic soil. We used MAP of 500 mm as a threshold of transition from arid to humid climate, and MAT of 10 °C as a threshold of transition from cool to warm climate.

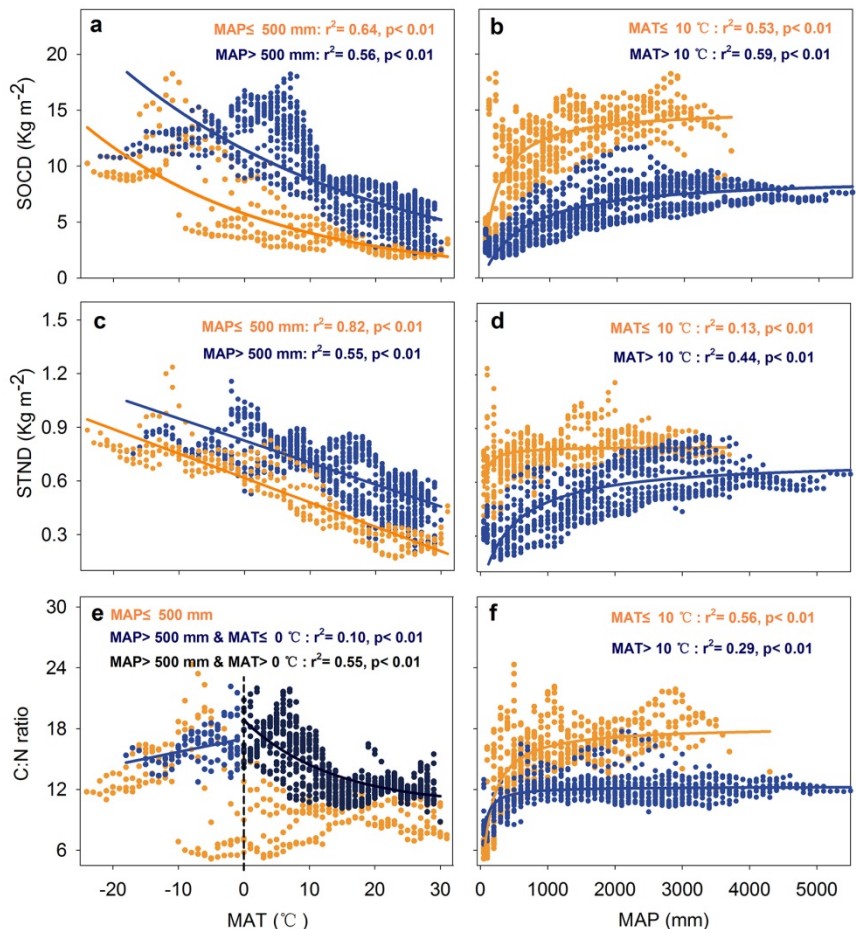

Figure 6. Changes in surface(0-30 cm) soil organic carbon density (SOCD), soil total nitrogen density (STND) and C:N ratios with mean annual precipitation (MAP) and mean annual temperature (MAT). We used MAP of 500 mm as a threshold of transition from arid to humid climate, and MAT of 10 °C as a threshold of transition from cool to warm climate. (a) and (b), SOCD; (c) and (d), STND; and (e) and (f), C:N ratios. Each dot shows the average value within each 1 °C MAT and 100 mm MAP.

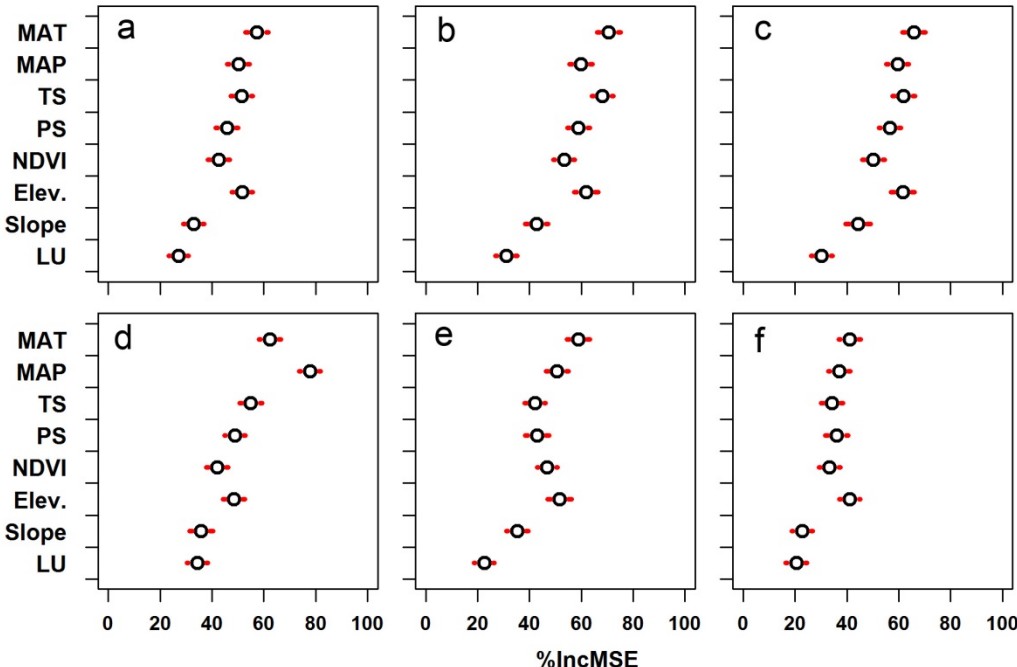

Figure 7. Importance of variables, denoted by the percent increase in mean-squared error (%IncMSE), for each soil property estimation. a, BD (bulk density, g m$^{-3}$); b, Sand fraction (%); c, Clay fraction (%); d, pH; e, SOCD (soil organic carbon density, kg m$^{-2}$); f, STND (soil total nitrogen density, kg m$^{-2}$). MAT, MAP, TS, PS, Elev. and LU indicate mean annual temperature, mean annual precipitation, annual temperature seasonality, annual precipitation seasonality, elevation and land use type, respectively.

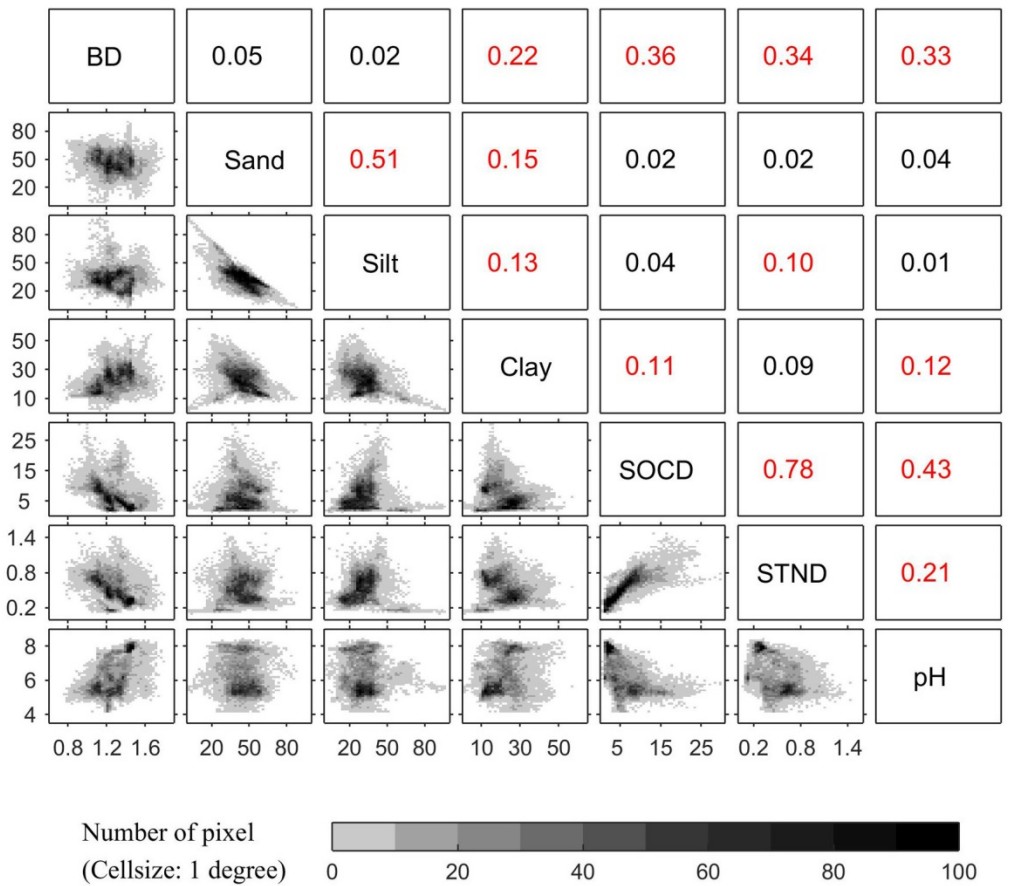

Figure 8. Correlation between surface (0-30cm) soil properties. $R^2$ between each two soil properties is shown in the upper plots with red color indicating $R^2 > 0.1$. BD, SOCD, and STND indicate bulk density, soil organic carbon density, and soil total nitrogen density, respectively.

**Table 1.** Mean values of surface (0-30 cm) soil properties across terrestrial biomes of the world.

| Biome | Area | Bulk density | Sand | Silt | Clay | pH | SOCD | STND | C:N ratio | SOC stock | STN stock |
|---|---|---|---|---|---|---|---|---|---|---|---|
| | ($10^6$ ha) | ($g \cdot m^{-3}$) | (%) | (%) | (%) | | ($kg \cdot m^{-2}$) | ($kg \cdot m^{-2}$) | | (Pg) | (Pg) |
| TroF | 1877 | 1.27±0.10 | 42.42±9.37 | 27.05±7.25 | 30.53±5.97 | 5.30±0.74 | 5.67±1.62 | 0.48±0.13 | 11.79±1.53 | 107±0.56 | 9±0.05 |
| TemF | 992 | 1.28±0.21 | 45.43±8.94 | 34.97±6.89 | 19.59±5.65 | 5.80±0.80 | 8.82±3.46 | 0.64±0.18 | 13.99±4.07 | 87±0.47 | 6±0.04 |
| BF | 1435 | 1.16±0.12 | 50.01±8.77 | 32.77±6.03 | 17.22±4.52 | 5.36±0.31 | 11.11±3.56 | 0.66±0.15 | 17.07±4.98 | 159±1.94 | 10±0.21 |
| TSG | 1915 | 1.34±0.10 | 47.92±13.06 | 25.64±16.68 | 26.44±9.05 | 6.25±0.80 | 3.82±1.65 | 0.32±0.13 | 12.28±4.05 | 73±0.62 | 6±0.04 |
| TGS | 1148 | 1.30±0.16 | 45.14±11.13 | 34.71±10.68 | 20.15±6.73 | 6.97±0.58 | 5.36±2.45 | 0.55±0.20 | 10.19±3.27 | 62±0.41 | 6±0.05 |
| Deserts | 2674 | 1.40±0.12 | 43.28±10.66 | 33.67±12.89 | 23.05±6.80 | 7.45±0.63 | 2.78±1.09 | 0.34±0.12 | 8.64±2.47 | 74±1.19 | 9±0.10 |
| Tundra | 644 | 1.16±0.16 | 47.57±8.92 | 37.02±7.35 | 15.41±3.12 | 5.44±0.34 | 13.78±4.51 | 0.81±0.19 | 17.18±5.06 | 89±2.07 | 5±0.10 |
| Croplands | 1984 | 1.34±0.15 | 40.70±11.08 | 33.13±9.87 | 26.16±7.04 | 6.40±0.74 | 6.54±2.80 | 0.58±0.22 | 11.36±2.19 | 130±0.50 | 11±0.09 |
| PW | 159 | 1.23±0.12 | 41.24±8.74 | 33.45±7.59 | 25.31±6.16 | 5.83±0.53 | 9.77±4.16 | 0.72±0.23 | 13.62±4.11 | 16±0.16 | 1±0.04 |
| Total | 12829 | 1.29±0.16 | 45.20±10.84 | 32.17±10.97 | 22.63±7.89 | 6.18±1.01 | 6.94±4.42 | 0.53±0.23 | 12.63±4.68 | 797±4.10 | 64±0.41 |

Notes: We include croplands and permanent wetlands in this table, although they are not single biomes. Abbreviations: TroF, Tropical forests; TemF, Temperate forests; BF, Boreal forests; TSG, Tropical savannahs and grasslands; TGS, Temperate grasslands and shrublands; PW, Permanent wetlands. Spatial variability of soil properties within each biome was estimated as standard deviations. Uncertainties of total SOC and STN stocks were estimated as standard deviations based on10-fold cross-validation.