# Peer review of "Global soil-climate-biome diagram: linking surface soil properties to climate and biota"

_Biogeosciences, 2018_

## Referee Comment (RC1) · Petr Capek (Referee) · 20 Mar 2019

General comments

Zhao and co-authors present very interesting world-wide statistical analysis of soil physical and chemical characteristics variability in respect to climatic drivers. They create "biome" plots and maps of soil properties. The selected statistical approach is very innovative, which makes the manuscript sound and worth publishing. I am very skeptical about the direct causality. This issue is, however, relatively well covered in the discussion section of the manuscript. Nevertheless, there is one issue that is not covered in the discussion section at all and that is very important according to my opinion. Authors completely ignore soil orders and associated soil horizons. They report data for top 30 cm of soil. These 30cm can contain either single horizon or several very different horizons with very different physical and chemical properties.

My major concern surrounds the results presentation. Authors need to provide more information about the various statistical analyses they used to make Figures 4 – 6 and they also need to clarify various threshold they defined. The manuscript often contains either very vague or very strong statements unsupported by the results (see specific comments). For this type of presented results I think it is especially important to present uncertainty in quantitative terms. Any potential user of the extrapolated maps information/database should be aware of the limitations.

Specific comments:

*Page 1,* Lines 16 – 17: What is the "critical MAP for the transition from alkaline to acidic soil"? This is some model parameter?

*Page 1*, Line 18: two dots

*Page 1*, Lines 18 – 19: I do not understand the meaning of the last sentence of the abstract. Can authors clarify its meaning?

*Page 1*, Lines 21 – 23: This is very vague statement that deserves more clarity.

*Page 1*, Lines 25 – 26: Again, the statement is very vague. I would suggest more specific statement.

*Page 1*, Lines 29 – 30: What doesn't mean "soil stewardship for societal well-being"

*Page 2*, Lines 25 – 26: Please check the superscripts.

*Page 2*, Line 30: Third and very important soil-forming factor is the bedrock. I think authors should mention it right away in the introduction, not only in discussion section.

*Page 3*, Line 20: Please specify the type of pedologic data that GSD contains. Way these data wasn't used in analysis?

*Page 4*, Line 4: Authors excluded 10% of all observations (those above and below 95% and 5% quantile respectively). Is there a specific reason for that? In the case of this dataset, it is very difficult to identify outliers. First 30 cm of soil can include one or several different soil horizons with very different physical and chemical properties. Values identified as outliers might be very likely correct and reflect the difference between different soil horizons of different soil orders. Quick look on Fig. 4a suggest to me that all peatlands were very likely removed from the dataset. For that reason I would strongly suggest to keep all data in the dataset.

*Page 4*, Line 18: Please check the superscript.

*Page 4*, Lines 24 - 25: Can authors clarify the uncertainty estimation of C to N ratio? The uncertainty of the ratio composed of two variables, each with its uncertainty, should be calculated differently than the uncertainty of a single variable.

*Page 4*, 2.4. Statistical analysis: Based on presented results, this section requires more detailed information. Most importantly, authors should clarify the statistics reported in figures 4 – 6. See also the specific comments bellow.

*Page 5*, Line 15: Can authors also report here calculated total amounts of SOC and STN (Tab. 1)? It would be very interesting to compare this estimate with previous estimates in respect to chosen statistical approach in the discussion section. The estimates reported in Tab. 1 doesn't seem to me very different from previous estimates. Does it mean that the approach selected by authors is not so different in terms of the outputs?

*Page 5*, Line 25: "soillayer"

*Page 5*, Line 26: The "saturation curve" is mentioned here for the first time. Why saturation curve, what does it mean and how it was calculated/estimated? All that should be thoroughly explained in the statistical analysis section. Also explain the term "saturation threshold".

*Page 5*, Line 29: Authors should also explain the difference between "cold" and "warm" climates. How was the temperature threshold defined? How the arbitrary selected threshold affects the results?

*Page 6*, Line 11 - 12: Again, this is very strong statement. Given the issues surrounding the true causality in the found relationships discussed in this section I would suggest to make a less strong statement.

*Page 6*, Line 30 - 31: The transition definition and calculation is not explained in statistical analysis section nor reported in results section. It is impossible to review these results without detailed explanation.

*Page 7*, Line 10: Please check the subscript.

*Page 7*, Line 12 - 20: C to N ratio does not represent very good proxy for substrate quality so the discussion is very speculative at this point.

*Page 7*, Line 21 - 24: This is very speculative. Isn't the correlation simple given by the fact that plant derived organic material always contain C and N no matter what the limitation is?

*Page 8*, 4.4 Uncertainties in mapping soil properties at the global scale: Can authors quantify the uncertainty? According to Tab. S4, statistical model explains sometimes less than 20% of variability. In addition, 10% of all data were removed. I believe that the uncertainty is very important to state unambiguously so any potential user of the database and maps knows the limitations.

*Page 8*, Line 28: "Ourregion-specific"

*Page 8 and 9*, Conclusions: Again, very strong statements unsupported by the analysis.

Tables:

I would suggest to show only $R^2$ in table S4. It would improve the table alignment and reading.

Figures:

Fig. 4: I have concerns regarding MAP and MAT thresholds. Authors should clarify their definition and use. Specifically, arctic have typically low MAP but because of low MAT, their soils are very often water saturated.

Fig. 5: Please explain the part c of the figure in statistical analysis section.

Fig. 7: Results presented in this figure also require more information. How it was calculated? Was the increase of explained variability by a specific explanatory variable compared to null or some standard model?

Fig. 8: I did not find any reference to this figure in the main text.

---

## Referee Comment (RC2) · Anonymous Referee #4 · 31 Mar 2019

The article by Zhao et al., present an interesting global dataset for some soil parameters, linking these properties with climate and biota. Nevertheless, there are sevral issues that should be clarified and disucssed in much more details. The mentioned databases report row data for soil profiles, while the authors use also some parameters which are derived from these data (e.g. SOC and SON stocks). How these data were derived and harmonized should be better explained, since in the paper they are used to derive the linkages betwen soil, climate and biota. For the soil profiles in the different databases, were used only the soil layers having all the necessary parameters usefull to calculate the stocks of C and N? I am referring in particular to Bulk density and rock fragments content. If not, how the authors were dealing with this fact? They were using pedotransfer functions to derive bulk density? And if rock fragments con-

tent was missing? Since these two parameters are affecting wery much the stock the authors should make an effort in explaining how the database were harmonized. The discussion is sometimes weak. For instance the authors fould a correlation between bulk density, MAT and MAP. Similarly the call variation in relation to MAT and MAP? The discussion on the observed differences between ecosystems is quite poor. Not so many recent references are considered for the discussion. The effect of the vegetation on the selected soil parameters should be better considered and discussed.

Specific comment: Page 3 Line 5-10: "Compiled". And what about harmonization of the data?

Page 3 Line 20-30: Since most of the soil profiles were collected a very different range of years, how the climate was related to the propoerties? What you mean with pedological information? The fact the soil profiles data are presented by horizons?

Page 5 line 20-25: Which is the meaning of providing a mean global value for SOC and SON?

Page 6 line 5: In brackets (MAT < 400 mm) is probably MAP rather than MAT?

Page 6 line 20-30: the fact that bulk density is affected by precipitation and temperature should be better discussed. SImilarly the increases in clay content in relation to MAT and MAP. How soil erosion affect the clay fraction? Is soil erosion selective for the clay? And Silt and Sand? An effect of the actual land use on bulk density should also be pointed out in the discussion.

Page 7 line 5-10: The fact that in the tropical area Clay and bulk density decrease with altitude how can be explained? Which is the meaning of this decrease?

Figure 2: SOC density box Looking at the SOC density it appear that there is quite a lot of C in the North Mediterranean area, which is usually quite poor in SOC due to the continuous use of the land for agricultrue since millennia. On the other side also the area covered by tropical primart forests in Africa (e.g. Congo basis) seems to be

relatively poor? How they authors can explain these facts?

Bulk density box How the authors explain the very high values of BD for the United states? Why the are so high compared to other regions. Apparently in the USA there are not so many differences in BD in relation to the different ecosystems (e.g. Forests vs. grassland vs cropland)

Table 1. The BD of cropland appear to be similar to those of savanna and grassland. How it can be explained? Suìimilarly, concerning the SOC stock how it can be explained that cropland have similar values of tropical forests?

---

## Author Comment (AC1) · 5 May 2019

**Reply to comments by Petr Capek (RC1)**

**General comments**

Zhao and co-authors present very interesting world-wide statistical analysis of soil physical and chemical characteristics variability in respect to climatic drivers. They create "biome" plots and maps of soil properties. The selected statistical approach is very innovative, which makes the manuscript sound and worth publishing. I am very skeptical about the direct causality. This issue is, however, relatively well covered in the discussion section of the manuscript. Nevertheless, there is one issue that is not covered in the discussion section at all and that is very important according to my opinion. Authors completely ignore soil orders and associated soil horizons. They report data for top 30 cm of soil. These 30cm can contain either single horizon or several very different horizons with very different physical and chemical properties. My major concern surrounds the results presentation. Authors need to provide more information about the various statistical analyses they used to make Figures 4 – 6 and they also need to clarify various threshold they defined. The manuscript often contains either very vague or very strong statements unsupported by the results (see specific comments). For this type of presented results I think it is especially important to present uncertainty in quantitative terms. Any potential user of the extrapolated maps information/database should be aware of the limitations.

**Response**: Thanks for your helpful comments. First, we fully agree that ignoring the role of soil orders and horizons might result in uncertainties. In different countries/regions, soil classification was originally reported based on several different soil classification systems, such as the Unified Soil Classification System, FAO system, USDA Soil Taxonomy, Russia Soil Classification system, Australian Soil Classification system, and Chinese Soil Classification System. These soil classification systems are based on different standards (Carter and Bentley, 2016) and it is difficult to harmonize them (Batjes et al., 2007) and thus to quantify the role of soil orders. Additional, it is the same case for data of soil horizons and we were not able to consider the role of soil horizons. In our database, soil depth was well documented, while some literature data (15% profiles) did not report horizon information. We thus estimated the soil properties by a fixed depth of 30 cm and the depth of 0-30 cm has been frequently used in the mapping and modelling of surface soil properties at regional and global scales (e.g., Batjes, 1997; Yang et al., 2010; Saiz et al., 2012; Wieder et al., 2013; Shangguan et al., 2014). In the revised manuscript we have discussed the uncertainties. Thanks for your understanding!

Second, more details on the statistical analysis and the climate thresholds have been included for Figures 4-6 in the Methods session as follows: "For Figures 4-6, we first averaged soil property values for each MAT×MAP combination by a division of 1°C×100mm. We then plotted soil properties with climate variables. Specifically, a MAP at 500 mm is used to indicate a threshold for arid climates, while a MAT at 10 °C is used to separate relatively warmer and colder climates. The MAP threshold was based on the diagram of Holdridge life zone which used <500 mm to indicate arid climates (Holdridge, 1967). The threshold MAT was based on a universal thermal scale that used mean monthly temperature (approximately MAT) 10 °C to differentiate "cool" and colder from "mild" and warmer climates (Trewartha and Horn, 1980).

Additionally, we have carefully revised our manuscript according to your suggestions. Please see more details in our reply to your specific comments.

**Specific comments:**

Page 1, Lines 16 – 17: What is the "critical MAP for the transition from alkaline to acidic soil"? This is some model parameter?

**Response**: This sentence has been modified as follows: "Our results show that soil pH decreases with increasing MAP. The 'critical MAP' here means the corresponding MAP at soil pH=7.0, which indicates a shift from alkaline to acidic soil".

Page 1, Line 18: two dots
**Response**: Corrected.

Page 1, Lines 18 – 19: I do not understand the meaning of the last sentence of the abstract. Can authors clarify its meaning?
**Response**: Thanks for your suggestion. Our soil-climate-biome diagram implies strong effects of climate and biota on soil properties. Here we mean that soil properties may shift under global climate warming and land cover change. We have revised this sentence accordingly.

Page 1, Lines 21 – 23: This is very vague statement that deserves more clarity.
**Response**: Thanks. We have revised the sentence as "As a critical component of the Earth system, soils influence many ecological processes which provide fundamental ecosystem services." The next sentences explain this statement by using specific examples. "Soil physical properties, such as bulk density and soil texture, are important for green water retention and the preservation of carbon (C) and nutrients (Hassink, 1997; Sposito et al., 1999; Castellano and Kaye, 2009; Stockmann et al., 2013), whereas soil chemical properties, such as soil acidity (pH), organic C, and nutrient contents, are essential regulators of nutrient availability and plant growth, further affecting biogeochemical cycles as well as vegetation-climate feedbacks (Davidson and Janssens, 2006; Chapin et al., 2009; Milne et al., 2015)".

Page 1, Lines 25 – 26: Again, the statement is very vague. I would suggest more specific statement.
**Response**: Thanks. We have revised the sentence as "soil chemical properties, such as soil acidity (pH), organic C, and nutrient contents, are essential regulators of nutrient availability and plant growth, further affecting C and nutrient cycling as well as vegetation-climate feedbacks".

Page 1, Lines 29 – 30: What doesn't mean "soil stewardship for societal well-being"
**Response**: We meant soil stewardship to secure soil function and thus sustainable ecosystem services for the human society. We have revised this sentence accordingly.

Page 2, Lines 25 – 26: Please check the superscripts.
**Response**: Thank you. Typo corrected.

Page 2, Line 30: Third and very important soil-forming factor is the bedrock. I think authors should mention it right away in the introduction, not only in discussion section.

**Response**: Thanks for your suggestion. We have now mentioned the role of bedrock in the introduction section as follows: "Although parent material (e.g., bedrock) also plays an important role after climate and vegetation, it generally affects soil formation at a relatively long time scale (Chesworth, 1973) and particularly in the subsoil (Gentsch N et al., 2018). However, surface soils are dynamic in time and likely interacting more instantly with climate and vegetation (Weil et al., 2016)".

Page 3, Line 20: Please specify the type of pedologic data that GSD contains. Why these data wasn't used in analysis?

**Response**: Thanks for the suggestion. Our data base includes pedologic information on soil orders and soil horizons of sampled soil profiles. However, soil types were originally reported based on several different soil classification systems, such as the Unified Soil Classification System, FAO system, USDA Soil Taxonomy, Russia Soil Classification system, Australian Soil Classification system, and Chinese Soil Classification System. These soil classification systems define soil orders by different standards (Carter and Bentley, 2016) and it is difficult to harmonize them (Batjes et al., 2007) and thus quantify the role of soil orders. In our database, soil depth was well documented, while some literature data (15% profile) did not report horizon information. Therefore, we were not able to consider the role of soil horizons and simply estimated the soil properties by a fixed depth of 30 cm. In fact, the depth of 0-30 cm has been frequently used in the mapping and modelling of surface soil properties (e.g., Batjes, 1997; Yang et al., 2010; Saiz et al., 2012; Wieder et al., 2013; Shangguan et al., 2014). By considering essential climate (mean annual temperature, mean annual precipitation, seasonality of air temperature, seasonality of precipitation), vegetation (mean annual normalized difference vegetation index, and land use type) and topography factors (elevation, slope) that are key to soil formation (Jenny, 1941), our regional Random Forest analysis is may partially constrain the uncertainties due to ignoring soil orders and associated soil horizons. In the revised manuscript we have discussed the uncertainties. Thanks for your understanding!

Page 4, Line 4: Authors excluded 10% of all observations (those above and below 95% and 5% quantile respectively). Is there a specific reason for that? In the case of this dataset, it is very difficult to identify outliers. First 30 cm of soil can include one or several different soil horizons with very different physical and chemical properties. Values identified as outliers might be very likely correct and reflect the difference between different soil horizons of different soil orders. Quick look on Fig. 4a suggest to me that all peatlands were very likely removed from the dataset. For that reason I would strongly suggest to keep all data in the dataset.

**Response**: Thanks for your suggestion. We are sorry that there is a misunderstanding for the proportion of data excluded as outliers. We understand that it is difficulty to identify outliers. Outliers were usually excluded based on a certain criteria to reduce noises and improve the accuracy of prediction (Pleijsier, 1989; Batjes et al., 2007; Jiménez-Muñoz et al., 2015). For

instance, two detection criteria are frequently used for outlier identification: 1) samples deviating more than three standard deviations from the mean (±3σ criterion, Jiménez-Muñoz et al., 2015); 2) samples falling outside the range above a certain upper quartile and/or below a certain lower quartile. The latter is useful when the dataset is not normally distributed. In our analysis, we divided the global land into 11 regions to overcome spatial biases of the database and samples above 97.5% and below 2.5% quantile were excluded in each region to obtain a robust variogram. In fact, we only excluded 5% of the data as outliers. We have mapped the excluded sites and found that these sites are distributed relatively random in space (Fig. R1). This implies that excluding the outliers doesn't bias the datasets. In the revised manuscript, we have included maps for these excluded sites of observation in the supplements.

As you have suggested, we have spent two weeks to conduct a reanalysis by using all data samples and compared the results with those excluding outliers. We found similar patterns and means of global surface soil properties based on all data samples (Fig. R2; Table R2), but the cross-validated $R^2$ was obviously decreased for SOCD and STND, especially in US and Russia (Table R1). This implies that excluding outliers can improve the accuracy of prediction. We thus present the results based on the analysis excluding outliers in the main text and will also include the results by using all data samples in the supplement. If the reviewer prefers to present the results based on all data samples in the revised main text, we will do so accordingly.

[Figure]

Fig R1. Distribution of global soil profiles. Outliers are plotted as a) green (<2.5%), and b)

red (>97.5%) circles (taking soil organic carbon data as an example).

[Figure]

Figure R2. Map of worldwide soil properties in the upper 30-cm soil layer based on analysis using whole datasets. a, BD (bulk density, g cm$^{-3}$); b, Sand fraction (%); c, Silt fraction (%); d, Clay fraction (%); e, pH; f, SOCD (soil organic carbon density, kg m$^{-2}$); g, STND (soil total nitrogen density, kg m$^{-2}$); and h, C:N ratio.

**Table R1** Coefficient of determination ($R^2$) of the Random Forest models.

| Region | 95% | | | | | | ALL DATA | | | | | |
|---|---|---|---|---|---|---|---|---|---|---|---|---|
| | Bulk density | Content of sand | Content of clay | pH | SOCD | STND | Bulk density | Content of sand | Content of clay | pH | SOCD | STND |
| Tropical Asia | 0.56 | 0.27 | 0.28 | 0.67 | 0.37 | 0.33 | 0.56 | 0.33 | 0.31 | 0.71 | 0.42 | 0.32 |
| Mexico | 0.55 | 0.39 | 0.41 | 0.63 | 0.55 | 0.49 | 0.52 | 0.41 | 0.45 | 0.67 | 0.52 | 0.48 |
| Africa | 0.50 | 0.44 | 0.41 | 0.64 | 0.512 | 0.50 | 0.44 | 0.49 | 0.43 | 0.64 | 0.58 | 0.43 |
| Continental US | 0.27 | 0.50 | 0.44 | 0.62 | 0.50 | 0.46 | 0.28 | 0.54 | 0.46 | 0.66 | 0.39 | 0.43 |
| Canada & Alaska | 0.33 | 0.43 | 0.49 | 0.56 | 0.46 | 0.40 | 0.37 | 0.48 | 0.53 | 0.59 | 0.41 | 0.39 |
| Russia | 0.28 | 0.21 | 0.39 | 0.48 | *0.29* | *0.10* | 0.24 | 0.25 | 0.44 | 0.47 | *0.16* | *0.06* |
| South America | 0.23 | 0.24 | 0.16 | 0.57 | 0.32 | 0.24/ | 0.28 | 0.22 | 0.19 | 0.6 | 0.3 | 0.17 |
| Europe | 0.29 | 0.20 | 0.36 | 0.49 | *0.32* | *0.20* | 0.23 | 0.25 | 0.37 | 0.53 | *0.15* | *0.08* |
| East Asia | 0.51 | 0.18 | 0.39 | 0.51 | 0.47 | 0.28 | 0.54 | 0.25 | 0.33 | 0.54 | 0.4 | 0.18 |
| Australia | 0.46 | 0.65 | 0.47 | 0.28 | 0.47 | 0.31 | 0.47 | 0.67 | 0.5 | 0.31 | 0.51 | 0.32 |
| West Asia | 0.55 | 0.31 | 0.41 | 0.62 | 0.58 | 0.36 | 0.49 | 0.3 | 0.37 | 0.66 | 0.51 | 0.41 |

Notes: Values are the averaged $R^2$ and RMSE from test dataset of 10-fold cross-validation.

**Table R2.** Mean values of surface (0-30 cm) soil properties by biome of the world based on analysis using whole datasets.

| Biome | Area (10⁶ ha) | Bulk density (g·cm⁻³) | Sand (%) | Silt (%) | Clay (%) | pH | SOCD (kg·m⁻²) | STND (kg·m⁻²) | C:N ratio | SOC stock (Pg) | STN stock (Pg) |
|---|---|---|---|---|---|---|---|---|---|---|---|
| TroF | 1877 | 1.27±0.10 | 42.26±9.39 | 27.39±7.69 | 30.36±6.11 | 5.29±0.73 | 5.71±1.66 | 0.48±0.13 | 11.83±1.73 | 107±0.56 | 9±0.05 |
| TemF | 992 | 1.28±0.21 | 45.47±9.27 | 35.08±7.15 | 19.45±5.62 | 5.82±0.81 | 8.84±3.48 | 0.64±0.19 | 14.09±4.33 | 88±0.47 | 6±0.04 |
| BF | 1435 | 1.16±0.13 | 49.82±9.03 | 33.07±6.15 | 17.11±4.57 | 5.39±0.32 | 11.09±3.60 | 0.67±0.16 | 16.90±5.21 | 159±1.94 | 10±0.21 |
| TSG | 1915 | 1.34±0.10 | 48.07±13.17 | 25.80±16.65 | 26.12±9.02 | 6.26±0.79 | 3.83±1.66 | 0.32±0.13 | 12.33±4.30 | 73±0.62 | 6±0.04 |
| TGS | 1148 | 1.30±0.16 | 45.48±10.87 | 34.51±10.22 | 20.01±6.72 | 7.01±0.59 | 5.33±2.47 | 0.55±0.20 | 10.14±3.43 | 61±0.41 | 6±0.05 |
| Deserts | 2674 | 1.40±0.12 | 43.89±10.49 | 33.22±12.87 | 22.89±7.27 | 7.50±0.63 | 2.80±1.11 | 0.33±0.13 | 9.00±3.14 | 75±1.19 | 9±0.10 |
| Tundra | 644 | 1.16±0.17 | 47.76±9.00 | 36.86±7.43 | 15.38±3.18 | 5.40±0.32 | 13.68±4.55 | 0.82±0.21 | 16.95±4.85 | 88±2.07 | 5±0.10 |
| Croplands | 1984 | 1.34±0.15 | 40.68±11.31 | 33.27±10.10 | 26.05±7.03 | 6.41±0.74 | 6.57±2.83 | 0.58±0.23 | 11.38±2.42 | 130±0.50 | 12±0.09 |
| PW | 159 | 1.23±0.12 | 40.61±9.32 | 34.25±8.16 | 25.15±6.20 | 5.85±0.53 | 9.81±4.16 | 0.73±0.24 | 13.58±4.37 | 16±0.16 | 1±0.04 |
| Total | 12829 | 1.29±0.17 | 45.29±10.89 | 32.24±10.99 | 22.47±7.96 | 6.20±1.02 | 6.95±4.42 | 0.53±0.23 | 12.67±4.77 | 797±4.10 | 64±0.41 |

Notes: We include croplands and permanent wetlands in this table, although they are not single biomes. Abbriations: TroF, Tropical forests; TemF, Temperate forests; BF, Boreal forests; TSG, Tropical savannahs and grasslands; TGS, Temperate grasslands and shrublands; PW, Permanent wetlands. Spatial variability of soil properties within each biome was estimated as standard deviations. Uncertainties of total SOC and STN stocks were estimated as standard deviations based on10-fold cross-validation.

Page 4, Line 18: Please check the superscript.

**Response**: Typo corrected. Thanks.

Page 4, Lines 24 - 25: Can authors clarify the uncertainty estimation of C to N ratio? The uncertainty of the ratio composed of two variables, each with its uncertainty, should be calculated differently than the uncertainty of a single variable.

**Response**: Uncertainties of SOCD and STND were both assessed based on the bootstrap method. Specifically, a robust estimate was derived by averaging the 10-fold cross-validation samples, and the uncertainty of the estimates was calculated as the standard deviation (SD) of the 10-fold cross-validation. We only reported the uncertainties of SOCD and STND in Figure S3 and we believe they could jointly indicate the uncertainty of soil C:N ratio, which was calculated based on predicted SOCD and STND. We have clarified this information in the revised manuscript.

Page 4, 2.4. Statistical analysis: Based on presented results, this section requires more detailed information. Most importantly, authors should clarify the statistics reported in figures 4 – 6. See also the specific comments bellow.

**Response**: Thanks for your suggestion. Regarding figures 4-6, we first averaged soil property values using a MAT division of 1°C and a MAP division of 100mm. To explore the roles of MAT and MAP as well as their interactions, we then plotted soil properties with climate variables (MAT/MAP) by roughly distinguishing climate types (humid vs arid; warm vs cold). Specifically, a MAP at 500 mm is used to indicate a threshold for arid climates, while a MAT at 10 °C is used to separate relatively warmer and colder climates (see more information in our reply to the general comments above). We have added this information in the revised manuscript.

Page 5, Line 15: Can authors also report here calculated total amounts of SOC and STN (Tab. 1)? It would be very interesting to compare this estimate with previous estimates in respect to chosen statistical approach in the discussion section. The estimates reported in Tab. 1 doesn't seem to me very different from previous estimates. Does it mean that the approach selected by authors is not so different in terms of the outputs?

**Response**: Thanks for your suggestion. We have reported total storage of SOC and STN in the revised text. We also compare these values with previous estimates in the discussion section. Generally, our results of global SOC and STN storage in surface soils are similar to previous estimates (797 vs 716 Pg, Scharlemann et al., 2014; 63 vs 63-67 Pg, Batjes, 1996). By using climate, vegetation, topography and land use variables as predictors, our region-specific RF approach likely produces more robust global maps of soil properties in a finer spatial resolution. Moreover, uncertainties of the prediction have also been estimated. The regional RF analysis has several advantages over traditional approach, such as the ability to model non-linear relationships, handle both categorical and continuous predictors, and resist overfitting and noise features (Breiman, 2001).

Page 5, Line 25: "soillayer"

**Response**: Typo corrected.

Page 5, Line 26: The "saturation curve" is mentioned here for the first time. Why saturation curve, what does it mean and how it was calculated/estimated? All that should be thoroughly explained in the statistical analysis section. Also explain the term "saturation threshold".

**Response**: Precipitation favors net primary productivity (Del Grosso et al. 2008) and thus the C inputs into the soil. Moreover, precipitation intensifies weathering of the parent material and increases soil acidification, thus increasing formation of SOC-stabilizing minerals (Chaplot et al., 2010; Doetterl et al., 2015) and reducing decomposition of soil organic matter (Meier and Leuschner, 2010). Our results thus indicate that SOCD increased with MAP, while it didn't exceed a certain threshold because of a constraint of C inputs (Del Grosso et al. 2008). Thus, we used a saturation curve to indicate the relationship between surface SOCD and MAP (cold climate: SOCD = 0.0737×MAP/(1+0.0049×MAP); warm climate: SOCD = 0.0144×MAP/(1+0.0016×MAP) ). Based on these curves, the saturation threshold for cold climate and warm climate were 14.5 kg C m$^{-2}$ and 8.0 kg C m$^{-2}$ (Fig R3), respectively. We have included more details in the revised manuscript.

[Figure]

Fig R3. Changes in upper 30-cm soil organic carbon density (SOCD) with mean annual precipitation (MAP). We used MAT of 10 °C as a threshold of transition from cool to warm climate. Each dot shows the average value within each 1 °C MAT and 100 mm MAP.

Page 5, Line 29: Authors should also explain the difference between "cold" and "warm" climates. How was the temperature threshold defined? How the arbitrary selected threshold affects the results?

**Response**: Many thanks for your comments. We have now included more details on the definition and uncertainties of the MAT thresholds. In the revised manuscript, we used a MAT at 10 °C to separate relatively warmer and colder climates. The threshold MAT was based on a universal thermal scale that used mean monthly temperature (approximately MAT) 10 °C to differentiate "cool" and colder climate from "mild" and warmer climate (Trewartha and Horn, 1980). We have tested the robustness of the trends by using different values of MAT (e.g., 8, 10 and 12°C) and found similar trends (Fig. R4; taking SOC, STN and C:N ratio as an example). We have revised the manuscript accordingly.

[Figure]

Figure R4. Trends in SOCD (a, b, c), STND (d, e, f) and C:N (g, h, i) ratio with MAP with different segmentations of MAT(8, 10, and 12 °C).

Page 6, Line 11 - 12: Again, this is very strong statement. Given the issues surrounding the true causality in the found relationships discussed in this section I would suggest to make a less strong statement.

**Response**: We have revised the sentence as "The soil-climate-biome diagram demonstrates the quantitative linkages between surface soil physical properties and climate variables at the global scale".

Page 6, Line 30 - 31: The transition definition and calculation is not explained in statistical analysis section nor reported in results section. It is impossible to review these results without detailed explanation.

**Response**: Thanks for your suggestion. Our results show that soil pH decreases with increasing MAP. The "critical MAP" here means the corresponding MAT at soil pH=7.0, which indicates a shift from alkaline to acidic soil. We then plotted the critical levels of MAP (100 mm division) at soil pH=7.0 with MAT. We have included these details in the revised statistical analysis section.

Page 7, Line 10: Please check the subscript.
**Response**: Typo corrected.

Page 7, Line 12 - 20: C to N ratio does not represent very good proxy for substrate quality so the discussion is very speculative at this point.

**Response**: Thanks for your comments. We agree that C:N ratio in soil organic matter is not a good proxy of substrate quality. Instead, C:N ratio in soil organic matter is a result of long-term litter decomposition. The biological decomposition of litter is mainly carried out by microbial decomposers, including bacteria and fungi, and their grazers, which have lower C:N values compared with most litter types. This creates a high N demand by decomposers while a considerable fraction of assimilated C is respired during decomposition. Based on a synthesis of long-term experimental results, Manzoni et al., (2008) demonstrated that the C:N ratio of the litter decreased throughout decomposition. Therefore, we now use C:N ratios of soil organic matter to indicate the degree of decomposition. We have also revised the discussion accordingly (Page 9, line 22-26).

Page 7, Line 21 - 24: This is very speculative. Isn't the correlation simple given by the fact that plant derived organic material always contain C and N no matter what the limitation is?
**Response**: Thanks for your comments. We have removed the speculative statements. We revised the sentence as "Previous meta-analyses indicate that C:N ratio in the soil is well-constrained at the global scale (Cleveland and Liptzin, 2007). Accordingly, our results indicate a strong correlation between STN density as SOC density (Fig. 8) and demonstrate a similar pattern of STN density as SOC density across space and climate regimes (Figs. 3 & 6)" (Page 9, line 20-25).

Page 8, 4.4 Uncertainties in mapping soil properties at the global scale: Can authors quantify the uncertainty? According to Tab. S4, statistical model explains sometimes less than 20% of variability. In addition, 10% of all data were removed. I believe that the uncertainty is very important to state unambiguously so any potential user of the database and maps knows the limitations.
**Response**: Thanks for your suggestions. Modelling uncertainty was calculated as the standard deviation (SD) of the 10-fold cross-validation. Figure R5 shows that uncertainties were relatively higher in regions with less recorded soil profiles, such as high-latitude Russia and Canada. Moreover, soil property data below 2.5% quantile and above 97.5% quantile were excluded as outliers and we have mended our description (see our reply above). Although the values of $R^2$ is acceptable in most cases, we have also mentioned that our approach sometimes explains less than 20% of the variability due to several potential reasons, including limited sample size for some regions, data quality of original data, and limited independent variables used for modelling analysis. This has been discussed in detail in the revised manuscript.

[Figure]

**Figure R5.** Spatial pattern of uncertainties (standard deviation, SD, n=10) of soil properties in the upper 30-cm layer estimated by 10-fold cross-validation. a: Bulk density (g·cm$^{-3}$); b: Sand (%); c: Clay (%); d: pH e: SOCD (kg·m$^{-2}$); f: STND (kg·m$^{-2}$).

Page 8, Line 28: "Ourregion-specific"
**Response**: Thanks. Typo corrected.

Page 8 and 9, Conclusions: Again, very strong statements unsupported by the analysis.
**Response**: We have revised the concluding paragraph as "By compiling a comprehensive global soil database, we mapped eight soil properties based on machine learning algorithms and assessed the quantitative linkages between soil properties, climate, and biota at the global scale. Our region-specific random forest model generated high-resolution (1km) predictions of soil properties, which can be potentially used as inputs for regional and global biogeochemical models. Our results also produced a global soil-climate-biome diagram, which indicates the quantitative linkages between soil, climate, and biomes. Given that significant changes in major soil properties may occur in view of global environmental change (Trumbore and Czimczik, 2008; Chapin et al., 2009; Todd-Brown et al., 2013; Luo et al., 2016, 2017), more efforts should be made to understand the dynamics of the global soil-climate-biome diagram".

Tables:
I would suggest to show only R2 in table S4. It would improve the table alignment and reading.
**Response**: Thanks for your suggestions. We would like to keep both because they are essential indicators of model prediction accuracy.

Figures:

Fig. 4: I have concerns regarding MAP and MAT thresholds. Authors should clarify their definition and use. Specifically, arctic have typically low MAP but because of low MAT, their soils are very often water saturated.

**Response**: We have now included more details on the definition and uncertainties of these thresholds. In the revised manuscript, we have now used a MAP at 500 mm to roughly indicate a threshold for humid and arid climates, and a MAT at 10 °C to separate relatively warmer and colder climates (see our reply above). The MAP threshold was based on the diagram of Holdridge life zone which used <500 mm to indicate arid climates (Holdridge, 1967). The threshold MAT was based on a universal thermal scale that used mean monthly temperature (approximately MAT) 10 °C to differentiate "cool" and colder from "mild" and warmer climates (Trewartha and Horn, 1980). These thresholds were defined at global scale and we understand that these thresholds are not precise in some cases because of interactions between temperature and precipitation. For instance, soils can be relatively wet in regions with low MAP if low MAT doesn't result in too much evaporation (e.g., arctic ecosystems). Nevertheless, using these thresholds can help us to demonstrate the interactions of MAT and MAP in affecting soil properties (Figs. 4, 5, 6 in the revised manuscript). Thanks for your understanding.

Fig. 5: Please explain the part c of the figure in statistical analysis section.

**Response**: Thanks for your comments. Our results show that soil pH decreases with increasing MAP. The "critical MAP" here means the corresponding MAT at soil pH=7.0, which indicates a shift from alkaline to acidic soil. To explore the role of MAT in regulating the critical MAP, we then plotted the critical levels of MAP (100 mm division) at soil pH=7.0 with MAT. We have described this in the revised section of statistical analysis.

Fig. 7: Results presented in this figure also require more information. How it was calculated? Was the increase of explained variability by a specific explanatory variable compared to null or some standard model?

**Response**: Thanks for your comments. We have added the more detailed description in the methods section. Figure 7 indicated the relative importance of variables, denoted by the percent increase in mean-squared error (%IncMSE), which is estimated based on a permuting out-of-bag (OOB) method (Strobl et al., 2009 a &b). For each tree of the random forest, we compared the prediction error on the OOB portion of the data (MSE for regression) with that after permuting each predictor variable. The difference are then averaged over all trees, and normalized by the standard deviation of the differences. The relative percent (mean/SD) increase in MSE as compared to the out-of-bag rate (with all variables intact) was used to indicate the relative importance of each variable (Breiman, 2001). The results indicate that climates (MAT, MAP, TS, PS) and elevation are the most important variables for the prediction of each soil properties.

Fig. 8: I did not find any reference to this figure in the main text.

**Response**: Thanks a lot. We have referred Figure 8 in the revised manuscript. For instance, "Our results demonstrate that STND as SOCD showed a similar pattern (Figs. 3, S6) and a tight link (Fig. 8) across the global scale".

**References**

Batjes, N.H. (1996) Total carbon and nitrogen in the soils of the world. European Journal of Soil Science, 47,151–163.

Batjes, N.H. (1997) A world dataset of derived soil properties by FAO–UNESCO soil unit for global modelling. Soil Use and Management, 13(1), 9-16.

Batjes, N.H., Al-Adamat, R., Bhattacharyya, T., et al. (2007) Preparation of consistent soil data sets for modelling purposes: Secondary SOTER data for four case study areas. Agriculture Ecosystems & Environment 122(1): 26-34.

Breiman, Leo. (2001) Random Forests. Machine Learning, 45(1): 5–32.

Carter, M., & Bentley, S. P. (2016) Soil properties and their correlations. John Wiley & Sons.

Castellano, M. J., & Kaye, J. P. (2009) Global within-site variance in soil solution nitrogen and hydraulic conductivity are correlated with clay content. Ecosystems, 12(8), 1343–1351.

Chapin III, S., McFarland, J., McGuire, D. A., Euskirchen, E. S., Ruess, R. W., and Kielland, K. (2009) The changing global carbon cycle: linking plant–soil carbon dynamics to global consequences. J. Ecol., 97(5), 840–850.

Chaplot, V., Bouahom, B., Valentin, C., (2010) Soil organic carbon stocks in Laos: spatial variations and controlling factors. Global Change Biology 16 (4), 1380–1393.

Chesworth, W. (1973) The parent rock effect in the genesis of soil. Geoderma, 10(3), 215-225.

Davidson, E. A., and Janssens, I. A. (2006). Temperature sensitivity of soil carbon decomposition and feedbacks to climate change. Nature, 440(7081), 165.

Del Grosso, S., Parton, W., Stohlgren, T., Zheng, D., Bachelet, D., Prince, S., ... & Olson, R. (2008) Global potential net primary production predicted from vegetation class, precipitation, and temperature. Ecology, 89(8), 2117-2126.

Doetterl, S., et al., (2015) Soil carbon storage controlled by interactions between geochemistry and climate. Nature Geoscience, 8 (10), 780–783.

Gentsch N, Wild B, Mikutta R, et al. (2018) Temperature response of permafrost soil carbon is attenuated by mineral protection. Global Change Biology 24:3401–3415.

Hassink, J. (1997) The capacity of soils to preserve organic C and N by their association with clay and silt particles. Plant Soil, 191(1), 77–87.

Holdridge, L.R. (1967) Life zone ecology. Tropical Science Center. San Jose, Costa Rica .

Manzoni, S., Jackson, R. B., Trofymow, J. A., & Porporato, A. (2008) The global stoichiometry of litter nitrogen mineralization. Science, 321(5889), 684-686.

Meier, I.C., Leuschner, C., 2010. Variation of soil and biomass carbon pools in beech forests across a precipitation gradient. Glob. Chang. Biol. 16 (3), 1035–1045.

Milne, E., Banwart, S. A., Noellemeyer, E., Abson, D. J., Ballabio, C., Bampa, F., and Black, H. (2015) Soil carbon, multiple benefits. Environ. Dev., 13, 33–38.

Pleijsier, 1989. Variability in soil data. In: J. Bouma and A.K. Bregt (Editors), Land Qualities in Space and Time. PUDOC, Wageningen, pp. 89-98.

Saiz, G., Bird, M. I., Domingues, T., Schrodt, F., Schwarz, M., Feldpausch, T. R., ... & Diallo, A. (2012). Variation in soil carbon stocks and their determinants across a precipitation gradient in W est A frica. Global change biology, 18(5), 1670-1683.

Scharlemann, J. P., Tanner, E. V., Hiederer, R., & Kapos, V. (2014) Global soil carbon: understanding and managing the largest terrestrial carbon pool. Carbon Management, 5(1), 81-91.

Shangguan, W., Dai, Y., Duan, Q., Liu, B., & Yuan, H. (2014). A global soil data set for earth system modeling. Journal of Advances in Modeling Earth Systems, 6(1), 249-263.

Sposito, G., Skipper, N. T., Sutton, R., Park, S. H., Soper, A. K., and Greathouse, J. A. 1999. Surface geochemistry of the clay minerals. P. Natl. Acad. Sci. U.S.A., 96(7), 3358–3364.

Stockmann, U., Adams, M. A., Crawford, J. W., Field, D. J., Henakaarchchi, N., Jenkins, M., and Wheeler, I. (2013). The knowns, known unknowns and unknowns of sequestration of soil organic carbon. Agr., Ecosyst. Environ., 164, 80–99.

Strobl, Carolin; James Malley; and Gerhard Tutz. (2009) b. An introduction to recursive partitioning: rationale, application, and characteristics of classification and regression trees, bagging, and random forests. Psychological Methods. 14(4): 323–348.

Strobl, Carolin; Torsten Hothorn; and Achim Zeileis. (2009) a. Party on! A new, conditional variable importance measure for random forests available in party package. The R Journal. 1(2): 14-17.

Trewartha GT, Horn LH (1980) Introduction to climate, 5th edn. McGraw Hill, New York, NY.

Weil, R. R., and Brady, N. C. (2016) The nature and properties of soils. Pearson Education, USA.

Wieder, W. R., Bonan, G. B., & Allison, S. D. (2013) Global soil carbon projections are improved by modelling microbial processes. Nature Climate Change, 3(10), 909-912.

Yang, Y., Fang, J., Ma, W., Smith, P., Mohammat, A., Wang, S., & Wang, W. E. I. (2010). Soil carbon stock and its changes in northern China's grasslands from 1980s to 2000s. Global Change Biology, 16(11), 3036-3047.

---

## Author Comment (AC2) · 5 May 2019

**Reply to comments by Anonymous Referee #4 (RC2)**

**Comment #1**: The article by Zhao et al., present an interesting global dataset for some soil parameters, linking these properties with climate and biota. Nevertheless, there are several issues that should be clarified and discussed in much more details. The mentioned databases report row data for soil profiles, while the authors use also some parameters which are derived from these data (e.g. SOC and SON stocks). How these data were derived and harmonized should be better explained, since in the paper they are used to derive the linkages between soil, climate and biota. For the soil profiles in the different databases, were used only the soil layers having all the necessary parameters useful to calculate the stocks of C and N? I am referring in particular to Bulk density and rock fragments content. If not, how the authors were dealing with this fact? They were using pedotransfer functions to derive bulk density? And if rock fragments content was missing? Since these two parameters are affecting very much the stock the authors should make an effort in explaining how the database were harmonized. The discussion is sometimes weak. For instance the authors found a correlation between bulk density, MAT and MAP. Similarly the all variation in relation to MAT and MAP? The discussion on the observed differences between ecosystems is quite poor. Not so many recent references are considered for the discussion. The effect of the vegetation on the selected soil parameters should be better considered and discussed.

**Response:** Thanks for your helpful comments. We have revised the manuscript according to your suggestions:

First, we have included more details on the method to compile our global soil database (also see our reply to comment #2). Specially, SOC/SON stocks were calculated based on bulk density and concentrations of SOC/SON. We directly calculated the stocks of SOC and SON when all the necessary parameters were available. In the case that bulk density was not measured and SOC content was reported, we made estimates of bulk density based on regional-specific pedotransfer functions (Yang et al. 2007; Abdelbaki, 2018) and further estimated SOC/SON stocks. We first established empirical relationship between bulk density and SOC content in each regions (Table R1) and further estimated bulk density based on measured SOC for the soil profiles with missing data for bulk density. Overall, there were 42% profiles with measured data on bulk density and 58% profiles with estimated data on bulk density. We agree that correction for rock fragment is important to estimate soil C stocks, but it remains a global challenge because existing databases usually contain limited information on gravel fractions than bulk density and SOC concentrations (Jandl et al., 2014). Nevertheless, the inclusion of gravel and roots > 2 mm has been evidenced to exert a relatively low impact on the calculation of SOC stocks in the surface soil layer (0-30 cm), mainly due to the fact that surface soil usually contains a low proportion of gravels (Saiz et al., 2012). Currently, we assumed no rock fragment or rock issue had been handled if it was not reported, but we might use the mean gravel fractions of each vegetation type or soil orders as a potential correction factor. Nevertheless, this approach might also result in new uncertainty if used at the global scale. We may conduct such an analysis to deal with the gravel issue if the reviewer support this idea. We have also discussed the uncertainty due to missing gravel information in the revised manuscript. Thanks for your understanding!

Second, we have improved the discussion section by 1) discussing the potential causes

for the correlations between soil physical properties (bulk density and soil texture) and climate (MAT and MAP), 2) discussing the shifts of soil properties across biomes and the interactions between soil and vegetation, and 3) including more recent references. Please find more details in our reply to comments #6, 7, 8, 10 and associated references.

Table R1 Empirical regression models for relationship between bulk density (BD, g cm$^{-3}$) and soil organic carbon content (SOC, %) for each region.

| Region | Model | $R^2$ | Num |
|---|---|---|---|
| Tropical Asia | $BD = 1.336e^{-0.054\ SOC}$ | 0.26 | 765 |
| Mexico | $BD = 1.380e^{-0.061\ SOC}$ | 0.63 | 1243 |
| Africa | $BD = 1.480e^{-0.073\ SOC}$ | 0.30 | 3770 |
| Continental US | $BD = -0.173\ln(SOC) + 1.382$ | 0.45 | 1239 |
| Canada | $BD = 1.507e^{-0.027\ SOC}$ | 0.20 | 163 |
| Russia | $BD = -0.222\ln(SOC) + 1.287$ | 0.59 | 777 |
| South America | $BD = -0.07\ln(SOC) + 1.233$ | 0.15 | 2105 |
| Europe | $BD = 1.4661e^{-0.041\ SOC}$ | 0.60 | 2391 |
| East Asia | $BD = 1.4719e^{-0.08\ SOC}$ | 0.35 | 634 |
| Australia | $BD = 1.3319e^{-0.062\ SOC}$ | 0.74 | 167 |

**Comment #2**: Specific comment: Page 3 Line 5-10: "Compiled". And what about harmonization of the data?

**Response:** Thanks for your reminder! We have included more details on the methods of data screening and compiling in the revised manuscript and supplement. Along with ground-truth soil profile data (Table S1), we have also derived general information of soil sampling (site location, sampling time, source of data), pedologic information on soil orders and the horizons of the sampled soil profiles, mean annual temperature (MAT), mean annual precipitation (MAP), seasonality of air temperature (TS, calculated as $100 \times SD_{monthly}/Mean_{monthly}$), seasonality of precipitation (PS), mean annual normalized difference vegetation index (NDVI), elevation (global digital elevation map [DEM]), slope, and land use type for each recorded site (Table R1). Specifically for each profile, we recorded data on the number of horizon, top and bottom depth, and values of soil physical properties (sand/silt/clay fraction [%], gravel content [>2mm, %], bulk density [g/cm$^3$]), and chemical properties (pH, organic carbon content [%]; and total nitrogen content [%]) (Table R2). Data harmonization was conducted by four steps:

First, we screened sampling and measurement approaches of each soil property and excluded data those were not comparable to others in methodology. For instance, geographic coordinate data were included only when WGS84 or a geographic coordinate system that could be converted to WGS84 projection was used; Soil texture data were included only when the internationally accepted particle size class were used (clay < 2 μm < silt < 50 μm < sand < 2000 μm). This allows us to construct a database of soil properties with comparable methodology.

Second, we excluded records with no measured data on the target soil depth (0-30cm). In

case that soil organic matter was measured instead of soil organic C, we used a Bemmelen index (0.58) to convert organic matter into organic C. If data of bulk density were not measured, we made estimates based on regional-specific pedotransfer functions. We first established empirical relationship between bulk density and SOC content and further estimated bulk density based on measured SOC in case data were missing for bulk density.

Third, we extracted data on soil properties of the 0-30cm soil depth based on their depth of occurrence in a profile. SOC (STN) density was calculated based on bulk density and contents of SOC (STN).

Finally, we excluded values of each soil property departure from the median at the 95% level-of-confidence according to Pleijsier (1989). The remaining data were used for statistical analyses in order to reduce the influence of outliers.

We have also revised the section on data set in the revised manuscript (Page 3&4, 2.1 Data set).

Table R2 Information recorded in GSD.

| Site Data | Horizon Data |
|---|---|
| **Profile ID[a]** | **Profile ID & Horizon Id[b]** |
| | |
| **General:** | **General:** |
| Source of data | Horizon number |
| Description of year | depth, top |
| Soil classification | depth, bottom |
| | |
| **Site location and information:** | **Physical attributes:** |
| Location (description, region/ country) | Sand/Silt/Clay fraction (%) |
| Latitude & Longitude | Gravel content (>2mm, %) |
| Climate (MAT & MAP) | Bulk density ($g/cm^3$) |
| Elevation/ slope/ aspect | |
| Parent material | **Chemical attributes:** |
| Land use | Organic carbon (%) |
| | Total Nitrogen (%) |
| | pH-$H_2O$ |

a. unique indentifier for profile in GSD. b. Unique reference number for horizon within a profile. c. sand, 2.0-0.05mm; silt, 0.05-0.002mm, and clay, <0.002mm.

**Comment #3**: Page 3 Line 20-30: Since most of the soil profiles were collected a very different range of years, how the climate was related to the properties? What you mean with pedological information? The fact the soil profiles data are presented by horizons?

**Response**: Thanks for your comments. First, we know that soil profile data were measured across a very different range of years, but we used multiple-year mean values of climate variables in our analysis because soil properties were formed by subjecting to a climate for a long term. As 96% of soil profiles in GSD were sampled during 1950 to 2000, we used multiple-year (1950-2010) averages of climatic variables from WorldClim database. Second,

our database includes pedological information on soil orders and soil horizons of sampled soil profiles. We calculated surface soil properties (0-30 cm) based on data for each horizon. We have extended the discussion accordingly in the revised manuscript (Page 4, Line 15-18).

**Comment #4**: Page 5 line 20-25: What is the meaning of providing a mean global value for SOC and SON?

**Response**: We realized that this sentence doesn't belong here because this paragraph presents results on spatial patterns of soil properties. In the revised manuscript, we have moved this sentence to the end paragraph of section 3.2, which demonstrated results of the density and stocks of SOC and STN at global scale.

**Comment #5**: Page 6 line 5: In brackets (MAT < 400 mm) is probably MAP rather than MAT?

**Response**: Typo corrected.

**Comment #6**: Page 6 line 20-30: the fact that bulk density is affected by precipitation and temperature should be better discussed. Similarly the increases in clay content in relation to MAT and MAP. How soil erosion affect the clay fraction? Is soil erosion selective for the clay? And Silt and Sand? An effect of the actual land use on bulk density should also be pointed out in the discussion.

**Response**: Thanks for your comments and suggestions. In the revised manuscript, we have discussed the effects of climate, soil erosion and land use on soil physical properties (e.g., bulk density and soil texture).

First, the increase of bulk density with higher MAT and lower MAP is likely due to an accompanying decrease of SOCD (Ruehlmann and Körschens, 2009), which is jointly regulated MAT and MAP (Fig. R1; see more discussion on the effect of climate on SOCD in section 4.3; Wiesmeier et al., 2019). Higher MAT and MAP can accelerate the rate of weathering (Jenny, 1941; Lal, 2018), thus resulting in lower sand fraction and higher soil clay fraction.

Second, previous studies indicate that silt is most sensitive to soil erosion, while sand is less mobile due to high weight and clay is protected by soil aggregates (Wischmeier and Mannering, 1969; Torry et al., 1997; Wang et al., 2013).

Third, the effect of land use is important at a local scale. For instance, a change of forest or grassland to croplands can significantly decrease SOCD and thus decrease soil bulk density, while reforestation generally increases SOCD and thus decreases soil bulk density (Don et al., 2011). However, our static mapping of global soil properties are not able to account for the effect of temporal land use change.

[Figure]

Figure R1. Changes in SOCD with MAT and MAP.

**Comment #7**: Page 7 line 5-10: The fact that in the tropical area Clay and bulk density decrease with altitude how can be explained? Which is the meaning of this decrease?

**Response**: Thanks for your suggestion. We have discussed the possible causes for the altitudinal trends of bulk density and clay, which is similar to the trends across latitudes. First, the decrease of clay fraction with higher altitude is likely due to 1) a younger soil age (Waite and Sack, 2011), 2) lower weathering rate under lower temperature (Grieve et al., 1990; Kramer and Chadwick, 2016), and 3) a downslope translocation of surface soil to lower altitude. Second, the decrease of bulk density with altitude is likely due to an increase in SOC retention (Fig. R2f), which mainly results from low rate of decomposition along with lower temperature (Grieve et al., 1990 ; Kramer and Chadwick, 2016).

[Figure]

Figure R2. Changes in surface soil properties with elevation in tropical regions. a: Bulk density (g·cm⁻³); b: Sand (%); c: Silt (%); d: Clay (%); e:Ph; f: SOCD (kg C·m⁻²); g: STND (kg N·m⁻²); h: C:N ratio.

**Comment #8**: Figure 2: SOC density box Looking at the SOC density it appear that there is quite a lot of C in the North Mediterranean area, which is usually quite poor in SOC due to the continuous use of the land for agricultrue since millennia. On the other side also the area covered by tropical primary forests in Africa (e.g. Congo basis) seems to be relatively poor? How they authors can explain these facts?

**Response**: Thanks for your comments. We have mapped the original records of SOCD on the map in North Mediterranean croplands and found similar results as the mapped values (Fig.

R3). As indicated by a meta-analysis, croplands have significantly lower SOCD as compared with local plantation, forest and grassland (Don et al., 2011). In the North Mediterranean region, an increase in the area of olive plantation and vineyard in last decades might have contributed to the relatively high values of SOCD (Parras-Alcántara et al., 2013). We have separately mapped SOCD for global croplands (Fig. R3) and the values of SOCD in North Mediterranean area were not as high as the impression by Figure 2 in the manuscript. This is likely visual illusion due to a mix of croplands with natural vegetation.

Due to fast turnover with rapid decomposition of organic matter, SOC content has been evidenced to be relatively poor in tropical forests (e.g., Congo and Amazon tropical forests) (Wang et al., 2018). Accordingly, previous mappings of SOCD have also shown relatively low values in tropical forests (Köchy et al., 2015; Jackson et al., 2017).

[Figure]

Figure. R3. Site records (a) and spatial variations (b) of SOCD in croplands.

**Comment #9**: Bulk density box How the authors explain the very high values of BD for the United states? Why they are so high compared to other regions. Apparently in the USA there are not so many differences in BD in relation to the different ecosystems (e.g. Forests vs. grassland vs cropland)

**Response**: Thanks for the comments. We have summarized the original records of bulk density for each 11 regions (Table R3). The results showed that mean regional bulk density was also relatively high in the continental United States. We have further summarized the

original records of bulk density for forests, grassland and cropland in the US. We also found that bulk density of forests, grassland and cropland didn't show much difference (Table R4). Overall, our mapping of bulk density is in consistent with the pattern based on raw data and is similar to previous mapping on global bulk density (Hengl et al., 2014; Shangguan et al., 2014 ).

Table R3 Mean values of sampled bulk density data for each region in the GSD.

| Region | Bulk density (g·cm$^{-3}$) | | |
| --- | --- | --- | --- |
| | Mean | SD | Number |
| Tropical Asia | 1.33 | 0.23 | 860 |
| Mexico | 1.22 | 0.26 | 316 |
| Africa | 1.37 | 0.16 | 3740 |
| Continental US | 1.57 | 0.22 | 9322 |
| Canada | 1.25 | 0.32 | 790 |
| Russia | 1.12 | 0.28 | 386 |
| South America | 1.21 | 0.19 | 1764 |
| Europe | 1.27 | 0.30 | 1527 |
| East Asia | 1.29 | 0.20 | 2762 |
| Australia | 1.12 | 0.27 | 162 |
| West Asia | 1.48 | 0.20 | 333 |
| Alaska | 1.07 | 0.35 | 79 |
| Total | 1.33 | 0.23 | 860 |

Table R4 Mean values of sampled bulk density data for each biomes in the Continental US.

| Continental US | Bulk density (g·cm$^{-3}$) | | |
| --- | --- | --- | --- |
| | Mean | SD | Number |
| Forest | 1.57 | 0.28 | 1588 |
| Shrub | 1.64 | 0.27 | 1096 |
| Grassland | 1.56 | 0.20 | 2560 |
| Cropland | 1.54 | 0.15 | 3084 |
| All* | 1.57 | 0.22 | 9322 |

Note: Mean measured bulk density was not shown for savanna, wetlands and sparse vegetation because of limited sample size (<100). However, these biomes were also used to calculated regional mean of all biomes.

**Comment #10**: Table 1. The BD of cropland appear to be similar to those of savanna and grassland. How it can be explained? Similarly, concerning the SOC stock how it can be explained that cropland have similar values of tropical forests?

**Response:** Thanks. Table 1 shows global means of soil property across biomes, while bulk density shows significant spatial variations within savanna and grasslands (Fig. R4a) as well as croplands (Fig. R4b). Generally, bulk density ranged from ~1.0 to ~1.7 g·cm$^{-3}$ in savanna and grasslands (Fig. R4a), and it ranged from ~1.1 to ~1.7 g·cm$^{-3}$ in croplands (Fig. R4b). Considering the large spatial variation in soil properties and limited overlap in spatial distribution, it is difficult to attribute reasons to the difference of global means between

croplands and other biomes. This is the same for the comparison of global mean SOCD between croplands and tropical forests (Fig. R5). SOCD in tropical forests generally ranged from 3 to 10 kg·m$^{-2}$, while it ranged from 2 to 12 kg·m$^{-2}$ in croplands. When comparing values at a same region (e.g., southeast Asia), SOCD is obviously lower in croplands than in tropical forests (compare Fig. 5a and Fig. 5b). This difference has been also evidenced by meta-analysis based on field observations (Don et al., 2011).

[Figure]

Figure. R4. Spatial variations of bulk density in (a) savanna and grassland, and (b) croplands.

[Figure]

Figure. R5. Spatial variations of SOCD in (a) tropical forests, and (b) croplands.

**Reference**

[revised manuscript text omitted]

Yang Y.H., Mohammat A., Feng J.M., Zhou R., Fang J.Y. (2007) Storage, patterns and environmental controls of soil organic carbon in China. Biogeochemistry, 84, 131–141.